# Redox signaling via the molecular chaperone BiP protects cells against endoplasmic reticulum-derived oxidative stress

Jie Wang[1], Kristeen A Pareja[1], Chris A Kaiser[2], Carolyn S Sevier[1]*

[1]Department of Molecular Medicine, Cornell University, Ithaca, United States;
[2]Department of Biology, Massachusetts Institute of Technology, Cambridge, United States

**Abstract** Oxidative protein folding in the endoplasmic reticulum (ER) has emerged as a potentially significant source of cellular reactive oxygen species (ROS). Recent studies suggest that levels of ROS generated as a byproduct of oxidative folding rival those produced by mitochondrial respiration. Mechanisms that protect cells against oxidant accumulation within the ER have begun to be elucidated yet many questions still remain regarding how cells prevent oxidant-induced damage from ER folding events. Here we report a new role for a central well-characterized player in ER homeostasis as a direct sensor of ER redox imbalance. Specifically we show that a conserved cysteine in the lumenal chaperone BiP is susceptible to oxidation by peroxide, and we demonstrate that oxidation of this conserved cysteine disrupts BiP's ATPase cycle. We propose that alteration of BiP activity upon oxidation helps cells cope with disruption to oxidative folding within the ER during oxidative stress.

*For correspondence: css224@cornell.edu

Competing interests: The authors declare that no competing interests exist.

## Introduction

A largely unrecognized yet significant source of intracellular reactive oxygen species (ROS) is oxidative folding within the lumen of the endoplasmic reticulum (ER). Ero1 is a primary enzymatic catalyst of biosynthetic disulfide bonds (*Araki and Inaba, 2012*; *Ramming and Appenzeller-Herzog, 2012*). A byproduct of Ero1 activity is peroxide, a ROS. Studies with recombinant Ero1 demonstrate that for every disulfide bond generated by Ero1 a peroxide molecule is formed (*Tu and Weissman, 2002*; *Gross et al., 2006*). Extrapolating from these in vitro results, it is estimated that Ero1 activity in living cells could generate up to 25% of the cellular ROS produced during protein synthesis (*Tu and Weissman, 2004*). Fluorescent probes that measure ROS in living cells indicate that the amount of ROS in the ER lumen exceeds the quantity of ROS within the mitochondria, a substantial and well-characterized source of intracellular ROS (*Malinouski et al., 2011*).

Recently, several systems that function to consume ROS in the ER lumen have been uncovered in mammalian cells. A peroxiredoxin (PRDX4) and two glutathione peroxidases (GPx7/8) have been demonstrated to reduce peroxide to water (*Tavender and Bulleid, 2010*; *Zito et al., 2010*; *Nguyen et al., 2011*). These enzymes enhance the efficiency of oxidative protein folding by coupling peroxide reduction with oxidation of the disulfide-bond forming enzyme PDI, ultimately converting peroxide generated as a byproduct of disulfide bond formation into additional nascent chain disulfides (*Tavender et al., 2010*; *Zito et al., 2010*; *Nguyen et al., 2011*). We expect these identified detoxification pathways operate within the ER alongside additional and more general systems conserved across eukaryotes that remain to be elucidated. Lower eukaryotes maintain robust Ero1-dependent protein oxidation pathways that produce ROS, however these organisms do not contain homologs of either mammalian PDRX4 or GPx7/8.

**eLife digest** The endoplasmic reticulum is the cellular compartment where approximately one third of the cell's proteins are made. Inside, chaperone molecules bind to newly made protein chains and help them to fold into the three-dimensional structure required for the protein to work correctly. A chaperone called Ero1 helps to facilitate this folding process by catalyzing a reaction that forms strong chemical bonds, which help stabilize the final protein structures. However, this help from Ero1 comes at a cost: forming a stabilizing bond this way also produces a peroxide molecule as a byproduct.

Peroxide is a 'reactive oxygen species': a chemical that can oxidize and damage proteins and DNA, which can potentially kill the cell. Three other enzymes in the endoplasmic reticulum can convert peroxide into water, to protect the cells from reactive oxygen species build-up. However, not all cells that use Ero1 have these other enzymes, suggesting that other pathways must exist to manage reactive oxygen species.

Wang et al. took advantage of yeast cells containing a hyperactive mutant version of the Ero1 enzyme to look for alternative detoxifying mechanisms that occur when the cell is stressed by an excess of reactive oxygen species. In these cells, Wang et al. observed that the high levels of reactive oxygen species caused part of a chaperone molecule called BiP to oxidize. This modification of BiP acts like a switch that the reactive oxygen species flip on. When activated by the reactive oxygen species, BiP enhances its activity as a folding molecular chaperone, keeping proteins apart. This is thought to allow BiP to minimize the protein misfolding that may otherwise occur in the wake of the damage caused by the building levels of peroxide. Wang et al. created a mutant BiP chaperone that mimics the oxidized form, and found that it also protects cells from the damage inflicted by the excess of reactive oxygen species.

Wang et al. propose that the BiP chaperone may be an important sensor of reactive oxygen species that changes its activity when these harmful chemicals are present and helps to protect the cell from damage. The success in mimicking the protective effects of oxidized BiP with a mutant BiP suggest that in the future one may be able to design small molecule drugs that bind to BiP to produce the activity of the modified form.

To discover systems that respond to redox imbalance within the ER, we capitalized on our prior identification of a key homeostatic feedback system that modulates the activity of Ero1 in disulfide bond formation (*Sevier et al., 2007*). Regulation of Ero1 activity normally serves to maintain redox homeostasis in the ER lumen and helps to moderate the amount of ROS produced by Ero1. Disruption of the Ero1 feedback loop (de-regulation of Ero1) is detrimental to the ER redox environment, resulting in severe redox imbalance in the ER and oxidative stress (*Sevier et al., 2007*). We took advantage of the ROS generated by overproduction of a de-regulated, constitutively active mutant of Ero1 (Ero1-C150A-C295A; hereafter referred to as Ero1*) to identify cellular systems activated in response to an overabundance of ER ROS.

Here we report that oxidative stress in the ER of yeast, created by Ero1* overproduction, results in direct modification by peroxide of a conserved cysteine in the ATP-binding site of the molecular chaperone BiP. We show that modification of BiP decouples the ATPase and peptide-binding activities of BiP, disrupting the normal allostery between these domains. We propose that oxidation of BiP during conditions of increased ER ROS both enhances the ability of BiP to bind polypeptides and prevents consumption of cellular ATP by hydrolysis. We suggest that this conserved ROS-sensing mechanism has evolved to protect against oxidative stress in the ER by minimizing protein aggregation and maintaining ER folding homeostasis during conditions of excess ROS.

## Results

### A cysteine-less BiP strain is more sensitive to oxidative stress

A search of ER-localized protein sequences for potentially redox-responsive amino acid(s) drew our attention to a cysteine residue highly conserved among BiP orthologs. This cysteine is localized within the ATP-binding pocket (*Figure 1*) and is the sole cysteine in yeast BiP (Kar2 Cys63). To probe the role

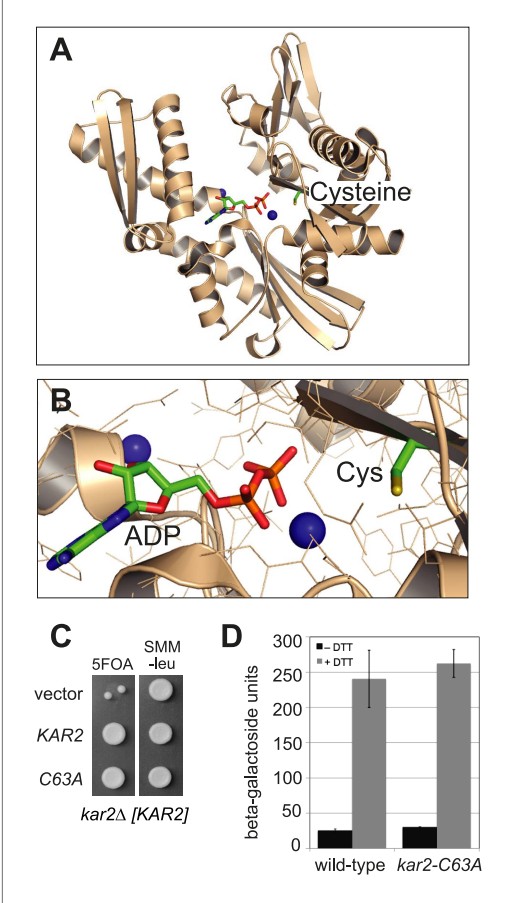

**Figure 1**. BiP contains a conserved cysteine that is dispensable for yeast viability. (**A**) Ribbon diagram of human BiP nucleotide binding domain in complex with calcium-ADP (PDB entry 3IUC) (*Wisniewska et al., 2010*). ADP and the conserved BiP cysteine are shown as colored sticks. Calcium atoms are represented as blue spheres. (**B**) Magnified representation of the conserved cysteine and CaADP from the BiP structure in panel **A**. Additional amino acid side chains are shown as lines. (**C**) CSY214 containing the plasmids pCS681, pCS685, or empty vector were spotted onto SMM plates with or without 5-fluoroorotic acid (5-FOA) and incubated for 2 d at 30°C. (**D**) CSY5 or CSY275 containing a UPRE-*lacZ* reporter plasmid (pJC8) were cultured in SMM-ura at 30°C, treated with 0 or 2 mM dithiothreitol (DTT) for 2 hr, and assayed for beta–galactosidase activity. Three independent transformants of each strain were grown and assayed in duplicate. Data represent the mean of averaged values for the three transformants ± SD.

of this conserved cysteine, we replaced the cysteine in yeast BiP with alanine and found that the cysteine is not required for the essential cellular function of BiP. Specifically, we observed that a *kar2-C63A* allele can complement an otherwise inviable *KAR2* chromosomal deletion strain (*kar2Δ*) (*Figure 1C*). Viability of the cysteine-less Kar2 strain was not a consequence of upregulation of the unfolded protein response (UPR), which has the potential to compensate for decreased Kar2 function through a transcription-mediated increase in Kar2 levels (*Beh and Rose, 1995*). As evidence, an UPR-promoter element (UPRE)-*lacZ* reporter showed a similar basal level of UPR induction for both wild-type and *kar2-C63A* strains (*Figure 1D*). BiP is not only a target of the UPR but also modulates UPR induction (*Pincus et al., 2010*). Replacement of the BiP cysteine with alanine did not perturb the UPR response; treatment with the reductant dithiothreitol (DTT), which disrupts disulfide bond formation and causes unfolded proteins to accumulate in the ER lumen, resulted in a comparable UPR induction in both wild-type and *kar2-C63A* strains (*Figure 1D*).

Although a *kar2-C63A* mutant showed no obvious defects in characteristic BiP activities, *kar2-C63A* cells were highly sensitive to hyper-oxidation of the ER lumen by overexpression of Ero1*. *kar2-C63A* strains containing either integrated (*Figure 2A*) or plasmid-borne (*Figure 2B*) *ERO1** alleles showed an increased sensitivity to hyper-oxidation of the ER by Ero1* overexpression relative to a wild-type strain. In *Figure 2A*, *ERO1** was integrated into the yeast genome (at the *CAN1* locus), allowing for stable and uniform expression of Ero1*. In *Figure 2B*, Ero1* expression was induced from a plasmid as described previously (*Sevier et al., 2007*). Importantly, overexpression of a catalytically inactive version of *ERO1** did not affect growth of the *kar2-C63A* strain relative to the wild-type strain (*Figure 2B*), confirming that the loss of viability upon Ero1* expression in the *kar2-C63A* mutant is a consequence of the hyper-oxidizing activity of Ero1*. As noted above, the *kar2-C63A* mutant alone (without Ero1* overexpression) grew at a rate similar to wild-type (*Figure 2A,B*).

Furthermore, the *kar2-C63A* mutant showed greater sensitivity than a wild-type strain to the small molecule oxidant diamide (*Figure 2C*), indicating a general role for the BiP cysteine in giving resistance to cellular oxidative stress. The *kar2-C63A* and wild-type strains showed similar growth in the presence of the reductant DTT (*Figure 2D*), suggesting that the conserved BiP cysteine has a selective role in managing oxidative but not reductive stress.

## BiP cysteine is oxidized by peroxide during oxidative ER stress

We speculated that oxidation of the BiP cysteine may trigger the ability of wild-type BiP to promote cell growth during hyper-oxidizing ER conditions. Accordingly, we used a biotin-switch assay to

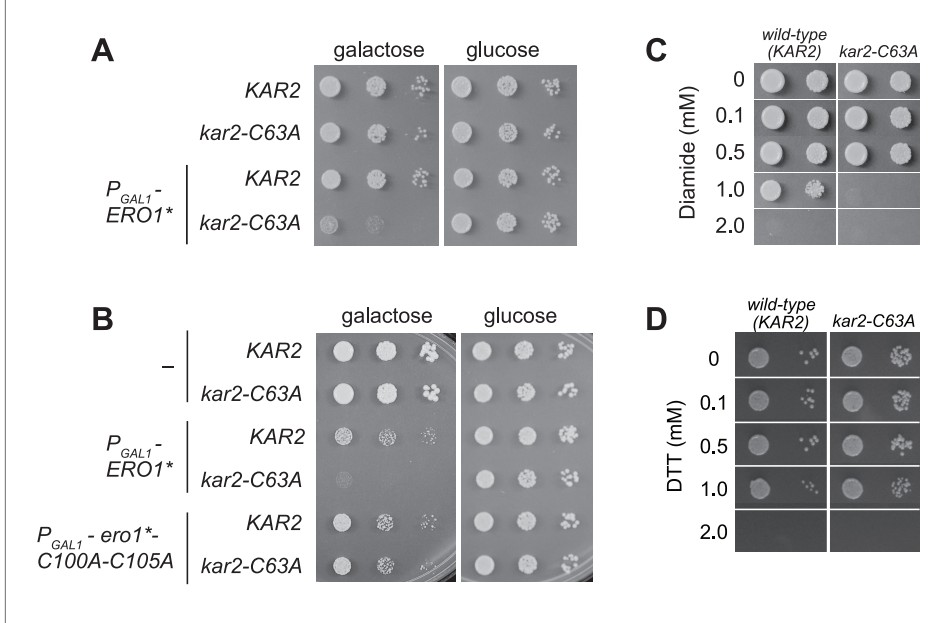

**Figure 2**. A cysteine-less BiP strain is sensitive to increased ER oxidation. (**A**) CSY170 and CSY278 strains containing an integrated galactose-inducible *ERO1\** were spotted onto SMM or SMM Gal plates, and plates were incubated for 2 d (glucose) or 3 d (galactose) at 30°C. (**B**) CSY5 or CSY275 strains transformed with plasmids pCS452, pCS504, or empty vector were spotted onto SMM-ura or SMM Gal-ura plates and incubated for 2 d (glucose) or 3 d (galactose) at 30°C. (**C** and **D**) CSY5 or CSY275 strains were spotted onto SMM plates containing 0–2 mM diamide (**C**) or DTT (**D**), and plates were incubated at 30°C for 2 d.

monitor the redox state of the BiP cysteine during oxidative stress (*Jaffrey and Snyder, 2001*; *Figure 3A*). Cells grown with or without Ero1\* induction were lysed under acidic conditions to protonate free thiols, which decreases thiol reactivity and limits post-lysis oxidation. Lysates were subsequently brought to neutral pH in the presence of N-ethylmaleimide (NEM), to block free thiols, followed by beta-mercaptoethanol (BME) treatment to reduce cysteines originally oxidized in the cell lysates. Cysteines originally oxidized in the cell, and subsequently reduced by BME, were then modified with maleimide-biotin. FLAG-tagged Kar2 was isolated from the lysates with anti-FLAG beads, separated by SDS-PAGE, and biotinylated Kar2 was detected by western blotting (*Figure 3B*).

Cells overexpressing Ero1\* exhibited an approximately twofold increase in the fraction of biotinylated Kar2 (*Figure 3B*, lanes 1 and 2), demonstrating a BME-reducible modification on Kar2 under hyper-oxidizing ER conditions. No Kar2-biotin signal was detected upon Ero1\* induction in a strain containing a cysteine-less variant of Kar2 (*Figure 3B*, lane 3), indicating biotinylation was limited to the Kar2 thiol. When lysates were not treated with BME prior to maleimide-biotin addition, a negligible biotin signal was detected for Kar2 (*Figure 3B*, lane 4), confirming that Kar2 biotinylation was not a consequence of residual Kar2 thiols available due to incomplete alkylation by NEM.

There are many known thiol-oxidation outcomes that are reversible by BME, including disulfide bonds, nitrosothiols, sulfenic acids, and glutathiolated cysteines (*Chung et al., 2013*). As a primary role of Ero1 is to facilitate disulfide bond formation in the ER, we suspected that Kar2 might form a disulfide bond with itself or another protein upon Ero1\* overproduction. However, we did not observe a mobility shift on a non-reducing SDS-polyacrylamide gel for Kar2 isolated from cells expressing Ero1\*, which would be expected if Kar2 was disulfide bonded to another protein (data not shown). Consequently, we focused on anticipated changes in small molecular oxidants in the ER that could modify Kar2. Specifically, as peroxide is a byproduct of Ero1\* activity, we considered oxidation of the BiP cysteine thiol (−SH) to sulfenic acid (−SOH).

To determine if peroxide oxidizes the Kar2 cysteine, we replaced the reductant BME in the biotin-switch assay with sodium arsenite, a reductant selective for sulfenic acid (*Torchinsky, 1991*). Treatment with sodium arsenite resulted in Kar2 biotinylation (*Figure 3C*, lane 1), demonstrating that Kar2

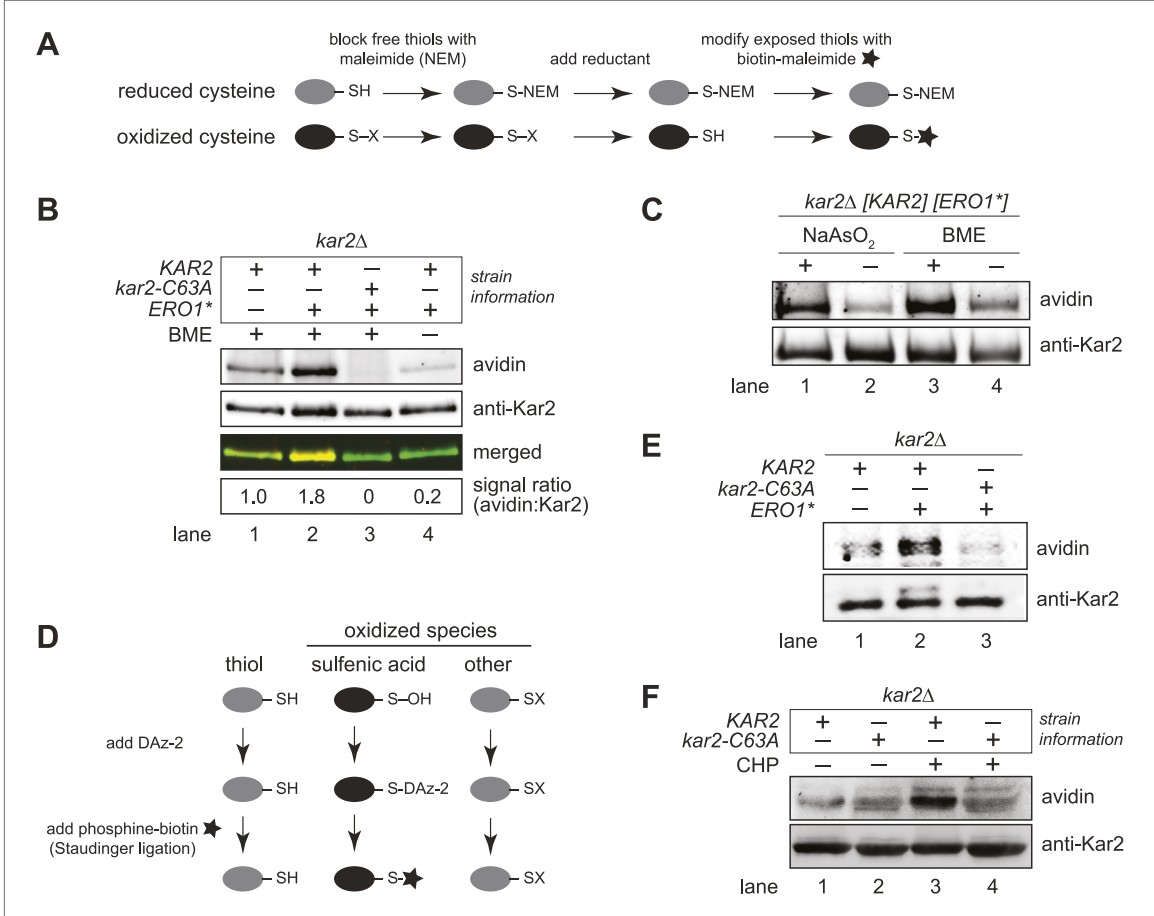

**Figure 3**. BiP's cysteine is oxidized upon hyperoxidation of the ER by Ero1*. (**A**) Schematic for the biotin-switch assay used in panels **B** and **C**. (**B**) The biotin-switch assay was used on lysates prepared from strains CSY316 and CSY319 containing either pCS452 or an empty vector. Strains were grown in galactose medium to induce Ero1*, and Kar2 was immunoprecipitated from lysates postbiotin-maleimide treatment. The relative proportion of Kar2 with an oxidized cysteine in the cell lysates under stressed (*ERO1*\*) and non-stressed conditions was determined by comparing the intensity of the Kar2-biotin signal relative to the total level of Kar2. As a control (lane 4), no reductant was added postNEM treatment and prior to biotin-maleimide addition. (**C**) Lysates were prepared from CSY316 containing pCS452 grown in galactose medium. The biotin-switch assay was performed with BME (as in panel **B**) or sodium arsenite (NaAsO₂) as the reductant. Kar2 was immunoprecipitated from lysates postbiotin-maleimide treatment. (**D**) Schematic for the use of DAz-2 to detect sulfenic acid in panel **E** and **F**. (**E**) Strains were grown as in panel **B** and treated with DAz-2. Kar2 was immunoprecipitated from lysates, and Staudinger ligation was performed with phosphine-biotin. DAz-2 addition was detected with an avidin probe. Note, a higher molecular weight Kar2 band corresponding to untranslocated Kar2 was observed in the Ero1*-overexpression strain (lane 2). Overlay of the two images confirmed that the avidin signal corresponds to the lower molecular weight mature Kar2 band (data not shown). (**F**) Strains were grown in glucose medium, and cells were exposed to 5 mM cumene hydroperoxide (CHP) for 30 min prior to harvest. Treatment with DAz-2 and sample processing were as described in panel **E**.

undergoes direct modification by peroxide. These data were confirmed using the dimedone-based DAz-2 probe, which binds directly to sulfenic acid-modified proteins (*Figure 3D*; *Leonard et al., 2009*). Kar2 proteins were isolated from cell lysates treated with DAz-2, Staudinger ligation was performed to add a biotin-moiety to any Kar2-DAz-2 species, and Kar2 modified by DAz-2-biotin was detected by western blotting with an avidin probe. An increase in biotinylated Kar2 was observed in DAz-2-treated cells overexpressing Ero1* relative to cells grown in the absence of stressor (*Figure 3E*), corroborating sulfenic acid formation at the site of the BiP cysteine during conditions of excess ER ROS. Oxidation of the Kar2 cysteine to sulfenic acid was detected also in cells exposed to exogenous peroxide. Addition of 5 mM cumene hydroperoxide (CHP) to cells for 30 min resulted in the recovery of biotinylated wild-type Kar2 from DAz-2 treated lysates, indicative of sulfenic acid formation on the Kar2 cysteine (*Figure 3F*, lane 3). Recovery of biotinylated-Kar2 was dependent on both the presence of the Kar2 cysteine and peroxide treatment (*Figure 3F*).

## Substitution of the conserved BiP cysteine with aspartic acid, phenylalanine, tyrosine, or tryptophan can protect against oxidative stress

Sensitivity of the *kar2-C63A* strain to Ero1* overexpression implies that BiP cysteine oxidation normally protects wild-type cells against oxidative cellular damage and a corresponding loss of viability during Ero1* overproduction. To provide more direct evidence for BiP oxidation in protection against oxidative ER stress, we created BiP alleles designed to mimic the modified form of BiP. Alleles were generated that contained a negatively charged or bulky amino acid in place of the BiP cysteine. We reasoned that the aromatic groups may recapitulate a structural perturbation caused by conversion of cysteine to sulfenic acid. (We will colloquially refer to these as bulky substitution alleles). Due to its charge, we anticipate aspartic acid may be a close mimetic of sulfenic acid, although it is likely an even better mimetic of sulfinic acid ($-SO_2H$), which is formed when sulfenic acid is further oxidized by peroxide.

When expressed ectopically, each of these alleles protected cells against a loss of viability during conditions of increased ER ROS, effectively mimicking the phenotype anticipated for oxidation of the BiP cysteine. Specifically, we found that addition of a plasmid coding for Kar2-C63D, Kar2-C63F, Kar2-C63Y, or Kar2-C63W suppressed the growth defect of a *kar2-C63A* strain overexpressing Ero1* (***Figure 4***). The growth of these strains could not be attributed solely to an excess of Kar2; addition of a Kar2-C63A expressing plasmid did not allow for growth of cells overexpressing Ero1* (***Figure 4***). The differences in growth for strains carrying the *kar2-C63D/F/Y/W* plasmids vs a *kar2-C63A* plasmid were most striking when cells were exposed to a combination of Ero1* and heat (37°C) (***Figure 4***); heat may induce an additional folding burden on the ER, which may serve to accentuate the growth differences between strains. Notably, cells with *kar2-C63F/Y/W* plasmids grew better than strains containing a

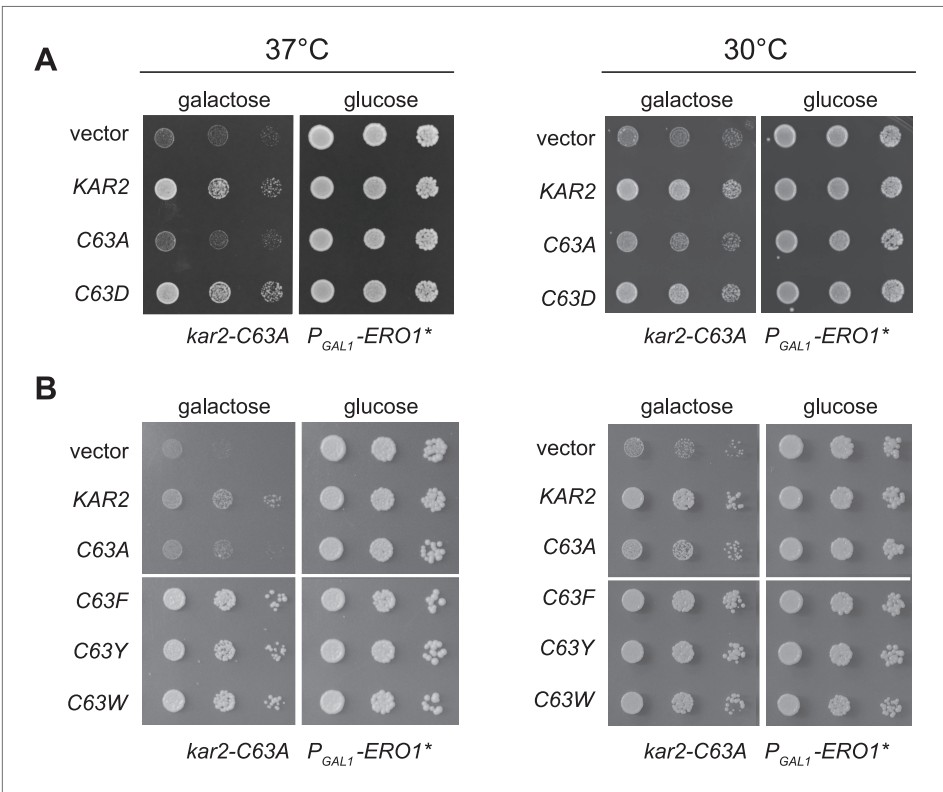

**Figure 4**. Substitution of the BiP cysteine with an amino acid containing a negatively charged or large side chain enables protection against hyper-oxidation of the ER lumen. (**A** and **B**) CSY278 containing (**A**) plasmids pCS681, pCS685, pCS802, or empty vector and (**B**) plasmids pCS681, pCS685, pCS687, pCS688, pCS750, or empty vector were spotted on SMM-leu or SMM Gal-leu plates, and plates were incubated at 30°C and 37°C for 2 d (glucose) or 3 d (galactose).

wild-type *KAR2* plasmid (*Figure 4*); these data may indicate an optimal ratio of unmodified-to-modified Kar2 that is closely matched by the combination of the genomic *kar2-C63A* and plasmid-borne *kar2-C63F/Y/W* alleles. The bulky alleles (Kar2-C63F/Y/W) were more protective than Kar2-C63D (*Figure 4*), suggesting that the Kar2-C63F/Y/W may be a more effective mimetic of the modified form than Kar2-C63D.

## Charged and bulky amino acid cysteine substitution alleles show decreased BiP activity in vivo and in vitro

Remarkably, although the negatively charged and bulky amino acid substitutions of Cys63 (Kar2-C63D/F/Y/W) protect against hyper-oxidizing conditions, these alleles lack essential BiP activity. *KAR2* is an essential gene in yeast. A Kar2-C63W allele could not support growth of a chromosomal deletion of *KAR2* (*Figure 5A*); Kar2-C63D, Kar2-C63F and Kar2-C63Y mutants were viable but temperature sensitive for growth (*Figure 5A,B*). BiP normally facilitates nascent chain translocation into the ER lumen. Decreased polypeptide translocation into the ER was observed for the Kar2-C63D, Kar2-C63F, and Kar2-C63Y strains after a 90 min shift to 37°C, further corroborating a loss of vital BiP activity for these cysteine substitution mutants (*Figure 5C*), Untranslocated polypeptides were readily detected by the accumulation of precursor protein forms (pre-Kar2, pre-PDI, Gas1 precursor) that lack the characteristic size shifts associated with post-translational processing that occurs after translocation into the ER lumen, including signal sequence processing, glycosylation (PDI, Gas1), and GPI anchor addition (Gas1) (*Figure 5C*). In keeping with a loss of vital BiP activity and disruption of the ER folding environment, an approximately fivefold induction of the UPR was observed in the Kar2-C63D/F/Y strains at both permissive (24°C) and restrictive (37°C) temperatures relative to wild-type cells (*Figure 5D*). UPR levels in the mutant strains at 37°C were comparable to those observed in cells treated with DTT,

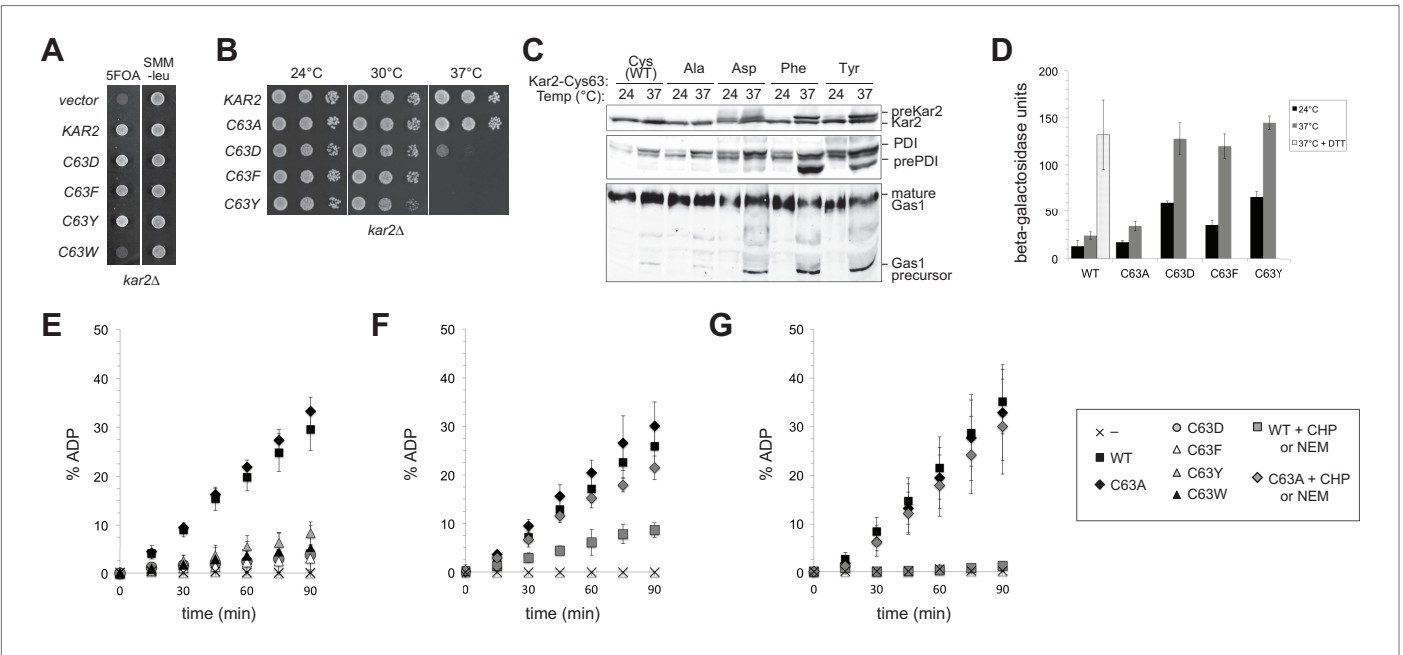

**Figure 5**. Replacement of the BiP cysteine with aspartic acid, phenylalanine, tyrosine, or tryptophan results in decreased BiP function. (**A**) CSY214 containing the plasmids pCS681, pCS802, pCS687, pCS688, pCS750 or empty vector were spotted onto SMM plates with or without 5-fluoroorotic acid (5-FOA) and incubated for 2 d at 30°C. (**B**) CSY289, 290, 368, 292, and 293 were spotted onto YPD plates and incubated at 24°, 30°, and 37°C. (**C**) Strains from **B** were cultured at 24°C to log-phase in YPD and shifted to 37°C for 90 min prior to harvest. Accumulation of unprocessed untranslocated forms of the proteins Kar2, PDI, and Gas1 were detected by western blotting. (**D**) Strains from **B** containing an UPRE-*lacZ* reporter plasmid (pJC8) were cultured in SMM-ura at 24°C to log-phase and shifted to 37°C (with or without 2 mM DTT) for 90 min prior to harvest. Samples were assayed for beta–galactosidase activity. Three independent transformants of each strain were grown and assayed in duplicate. Data represent the mean of averaged values for the three transformants ± SD. (**E–G**) ATP hydrolysis was assessed by determining the fraction of [alpha-$^{32}$P]ATP converted to [alpha-$^{32}$P]ADP as described in the 'Materials and methods'. Panels **F** and **G** show CHP and NEM-treated samples, respectively. Samples without chemical additions in panels **F** and **G** were mock treated to match the CHP or NEM-treatment. Data represent the means ± SD of three independent assays.

an established robust inducer of the UPR (*Figure 5D*). Significantly, the phenotypes observed for Kar2-C63D, Kar2-C63F, and Kar2-C63Y alleles contrast markedly the Kar2-C63A mutant, which showed wild-type activity with respect to growth, protein translocation, and UPR activity (*Figure 5*) and afforded no protection against oxidative stress (*Figure 4*). Note the ability of the loss-of-function Kar2-C63D/F/Y/W alleles to rescue the cell death phenotype from Ero1* overexpression (*Figure 4*) was observed at 37°C, the temperature at which none of these alleles are viable as the sole copy of cellular BiP.

The conserved BiP cysteine is located in the ATPase domain (*Figure 1*), and we speculated that the decrease in cellular BiP function observed upon introduction of a charged or bulky side chain at the cysteine position was a byproduct of altered BiP ATPase activity. To monitor ATPase activity, we expressed wild-type and BiP cysteine mutants in *Escherichia coli* and assayed the purified recombinant proteins *in vitro*. As anticipated, Kar2-C63D, Kar2-C63F, Kar2-C63Y, and Kar2-C63W strains all exhibited negligible hydrolysis of ATP (*Figure 5E*), whereas Kar2-C63A showed ATP hydrolysis activity equivalent to that observed for wild-type Kar2 (*Figure 5E*).

To correlate the bulky substitution BiP mutants with adduct formation at the BiP cysteine, we followed the ATPase activity of BiP after oxidation with peroxide (CHP; *Figure 5F*) or alkylation with NEM (*Figure 5G*). Oxidized or alkylated BiP exhibited a loss of ATPase activity similar to that observed for the negative or bulky amino acid cysteine-substitution mutants (*Figure 5E–G*). Treatment of Kar2-C63A with peroxide or NEM did not decrease ATP hydrolysis, confirming that the effect of peroxide and NEM was linked to cysteine modification (*Figure 5F,G*).

## Ero1* overexpression induces a blockade in translocation

The block in polypeptide translocation into the ER observed in cells expressing the BiP oxidation mimetic alleles raised the interesting possibility that oxidation of the BiP cysteine may trigger a block in polypeptide movement into the ER lumen. To assess whether oxidation of BiP impedes polypeptide translocation, we checked if cells overexpressing Ero1* accumulate unprocessed untranslocated precursor proteins, which show characteristic size shifts on SDS-PAGE. We observed that Ero1* overproduction results in an inhibition of protein translocation, which was most striking when Ero1* was overproduced in a sensitized strain background (an *ire1Δ* strain) that is unable to induce the UPR (*Figure 6*). A robust translocation block was not as reproducible in wild-type (*IRE1⁺*) cells, yet we did observe precursor protein accumulation in some assays using a wild-type strain background, which was dependent both on Ero1* overexpression and the presence of the Kar2 cysteine (e.g., *Figure 3E*, pre-Kar2 accumulation). We speculate the *ire1Δ* background facilitates a robust translocation block because there is no transcriptional upregulation of *KAR2* upon Ero1*-induction (due to the loss of a UPR response) (*Sevier et al., 2007*). Lower levels of BiP in the *ire1Δ* strain may result in a higher fraction of modified BiP upon Ero1* expression, which may lead to a more dramatic impact on ER translocation.

As noted above, it is intriguing that the translocation block correlated with conditions (Ero1* overproduction) that result in BiP modification (*Figure 3*). It is also notable that the sporadically observed translocation block in *IRE1+* cells expressing Ero1* is dependent on the presence of the Kar2 cysteine (e.g., *Figure 3E*, pre-Kar2 accumulation). Together these data suggest BiP modification may account for the observed translocation attenuation; however, at present we are unable to definitively link BiP oxidation with a block in polypeptide translocation. Attempts to facilitate a robust translocation block in wild-type cells with exogenous oxidant (diamide or CHP),

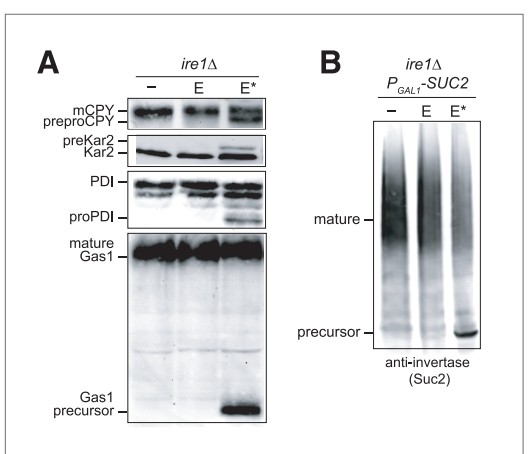

**Figure 6**. Overexpression of Ero1* causes an accumulation of untranslocated polypeptides. (**A**) CSY44 and (**B**) CSY172 containing plasmids pAF112 (*ERO1*; E), pCS452 (*ERO1**; E*), or empty vector were cultured in galactose medium for 5 hr to induce Ero1 and Suc2 expression. Accumulation of unprocessed untranslocated forms of the proteins CPY, Kar2, PDI, Gas1, and Suc2 were detected by western blotting.

or a combination of oxidant (Ero1*, diamide, CHP) and heat (37°C), have been unsuccessful; none of these conditions stimulate a robust reproducible translocation attenuation in wild-type cells. Experiments to study the role of the BiP cysteine in an *ire1Δ* background, where we do observe a strong translocation attenuation, have also been uninformative. To test the role of the BiP cysteine, we constructed a *kar2-C63A ire1Δ* double mutant overexpressing Ero1*, which would be expected to show a less pronounced translocation block if BiP oxidation triggers translocation attenuation. Although viable, a *kar2-C63A ire1Δ* double mutant exhibits heterogeneous colony size and slow growth, and experiments to study translocation attenuation during oxidative stress with this strain have produced inconsistent results.

## An ATPase-deficient BiP is not sufficient to protect cells against oxidative stress

Given the correlation between alleles of BiP that show a loss of ATPase activity and protection against oxidative stress, we asked whether ectopic addition of any BiP ATPase mutant could confer protection against over-oxidation of the ER. Mutation of a conserved threonine in the BiP ATPase domain to glycine has been established in both yeast (T249) and hamster BiP (T229) to prevent ATP hydrolysis (*Wei et al., 1995*; *Steel et al., 2004*). We expressed and purified recombinant Kar2-T249G and confirmed this mutant exhibits a severe defect in ATP hydrolysis relative to wild-type Kar2 (*Figure 7A*). Strikingly, ectopic expression of a Kar2-T249G mutant was not able to suppress the growth defect of a *kar2-C63A* strain overexpressing Ero1* (*Figure 7B*). These data suggest that decreased ATP hydrolysis by BiP, which we observed upon BiP oxidation, is not sufficient to allow for protection against redox imbalance in the ER lumen. These data imply that Kar2-C63D/F/Y/W mutants alter BiP activity in a manner distinct from Kar2-T249G.

## UPR induction by an ATPase-deficient BiP is not sufficient to manage oxidative stress

The robust UPR induction we observed in cells containing Kar2-C63D/F/Y alleles as the sole copy of BiP (*Figure 5D*) raised the possibility that these ATPase deficient BiP alleles might also elicit a UPR response when introduced into the *kar2-C63A* background. We reasoned UPR induction by the ATPase deficient BiP alleles could facilitate the protection conferred by ectopic expression of these Kar2 mutants during oxidative stress (*Figure 4*). Notably, introduction of a plasmid-borne copy of a Kar2-C63D/F/Y/W allele to *kar2-C63A* cells did result in an enhanced UPR relative to cells expressing wild-type Kar2 (or a Kar2-C63A mutant), primarily observed at high temperature (37°C) (*Figure 7C*). These data suggest UPR induction may contribute to the enhanced growth observed for these same strains under oxidative stress (*Figure 4*), and these data imply BiP oxidation during stress may trigger a UPR response. However, although UPR induction may augment growth during oxidative stress, UPR induction does not appear sufficient to manage the disruption to the ER environment in Ero1* overexpressing cells. Ectopic expression of the Kar2-T249G mutant in the *kar2-C63A* background (which does not facilitate growth during oxidative stress) resulted in a similar UPR as observed for the Kar2-C63D/F/Y/W mutants (*Figure 7C*).

## Modified BiP retains its ability to prevent protein aggregation

Recent data show that it is possible to decouple the normal allostery observed between Hsp70 ATPase and peptide binding activities. The small molecule YM-08 (an analog of the Hsp70 inhibitor MKT-077) has been shown to enhance Hsp70 binding to misfolded proteins and simultaneously inhibit Hsp70 ATPase activity (*Miyata et al., 2013*). Similarly, several mutations in the bacterial Hsp70 DnaK have been reported that allow DnaK to prevent protein aggregation in the absence of robust ATPase activity (*Chang et al., 2010*). These data raised the interesting possibility that oxidation of BiP could be a physiological means to allosterically decouple BiP activity, allowing BiP to prevent protein aggregation during excess ER ROS despite minimal ATPase activity.

To test whether modified BiP maintains the capacity to lessen protein aggregation, we employed an assay based on the in vitro aggregation of rhodanese. When a solution of reduced and chemically denatured rhodanese is rapidly diluted out of denaturant, rhodanese forms large aggregates that can be detected by light scattering (*Langer et al., 1992*; *Figure 8*). Significantly, we observed that the presence of the ATPase-deficient Kar2 cysteine-substitution alleles (Kar2-C63D/F/Y/W) during dilution not only lessened aggregation (minimized light scattering) of denatured rhodanese but also was more effective than wild-type BiP at minimizing rhodanese aggregation (*Figure 8A*). An enhanced capacity

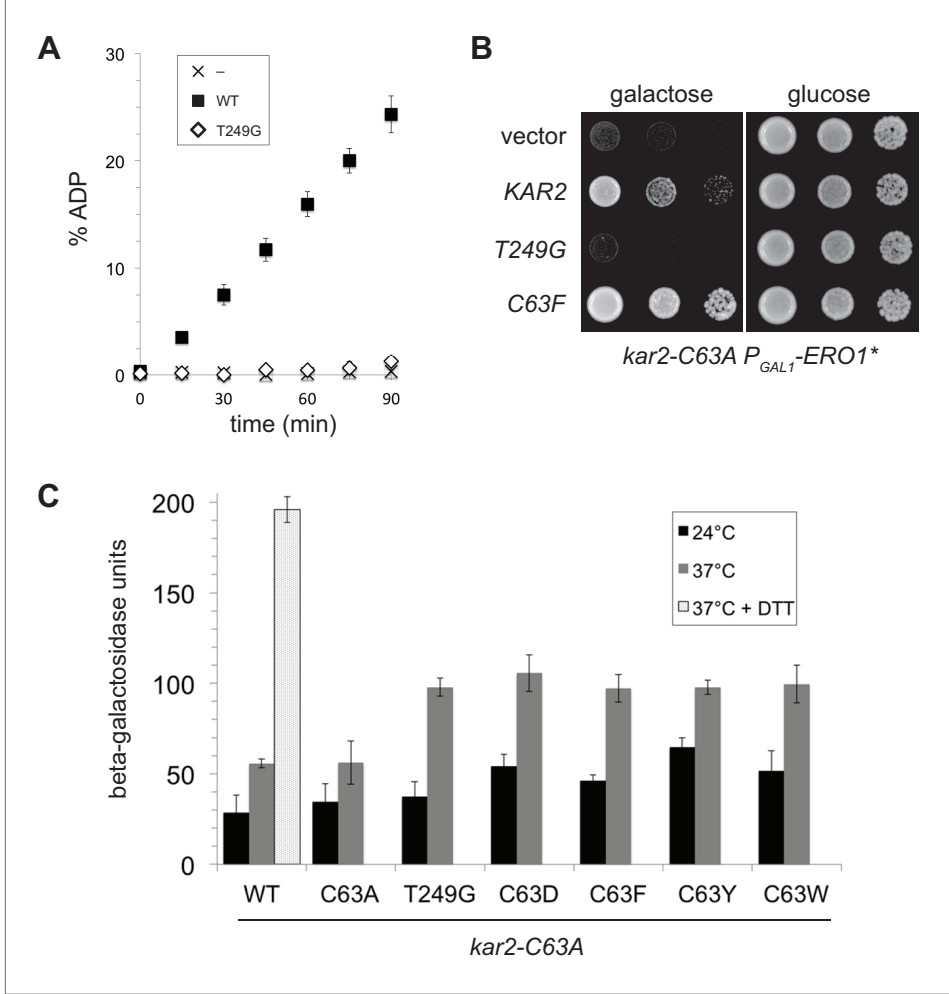

**Figure 7**. A BiP ATPase mutant is not sufficient to protect cells during oxidative stress. (**A**) ATP hydrolysis was assessed by determining the fraction of [alpha-$^{32}$P]ATP converted to [alpha-$^{32}$P]ADP as described in the 'Materials and methods'. Data represent the means ± SD of three independent assays. (**B**) CSY278 containing plasmids pCS681, pCS774, pCS687, or empty vector were spotted on SMM-leu or SMM Gal-leu plates, and plates were incubated at 37°C for 3 d. (**C**) CSY275 containing a UPRE-*lacZ* reporter (pCS852) and plasmids pCS681, pCS802, pCS774, pCS687, pCS688, or pCS750 were cultured in SMM-ura-leu at 24°C to log-phase and shifted to 37°C (with or without 2 mM DTT) for 90 min prior to harvest. Three independent transformants of each strain were grown and assayed for beta–galactosidase activity in duplicate. Data represent the mean of averaged values for the three transformants ± SD.

to prevent aggregation relative to wild-type BiP was observed as well for oxidized BiP (peroxide treated; Kar2-SOH) and alkylated BiP (Kar2-NEM) (*Figure 8B,C*). Treatment of a Kar2-C63A mutant with peroxide or NEM did not enhance its ability to lessen aggregation of rhodanese, demonstrating that the enhanced capacity to limit aggregation correlates with modification of the BiP cysteine (*Figure 8B,C*). Notably, the ATPase-deficient Kar2-T249G mutant (which did not enhance growth of cells during oxidative stress; *Figure 7*) did not show an enhanced ability to prevent aggregation of rhodanese (*Figure 8A*), supporting a correlation between a greater holdase capacity of BiP and enhanced viability during oxidative stress. It is worth pointing out that at the concentrations used in our assay, addition of wild-type, Kar2-C63A, and Kar2-T249G mutants did not allow for significant aggregation protection; the presence of any of these three proteins at the time of dilution resulted in light scattering comparable to the light scattering observed by rhodanese diluted into a solution of bovine serum albumin (BSA; *Figure 8A*). Assays with a second model substrate, IgY, showed similar trends. Kar2-C63F/Y and alkylated BiP (Kar2-NEM) showed a greater ability to curtail IgY aggregation than

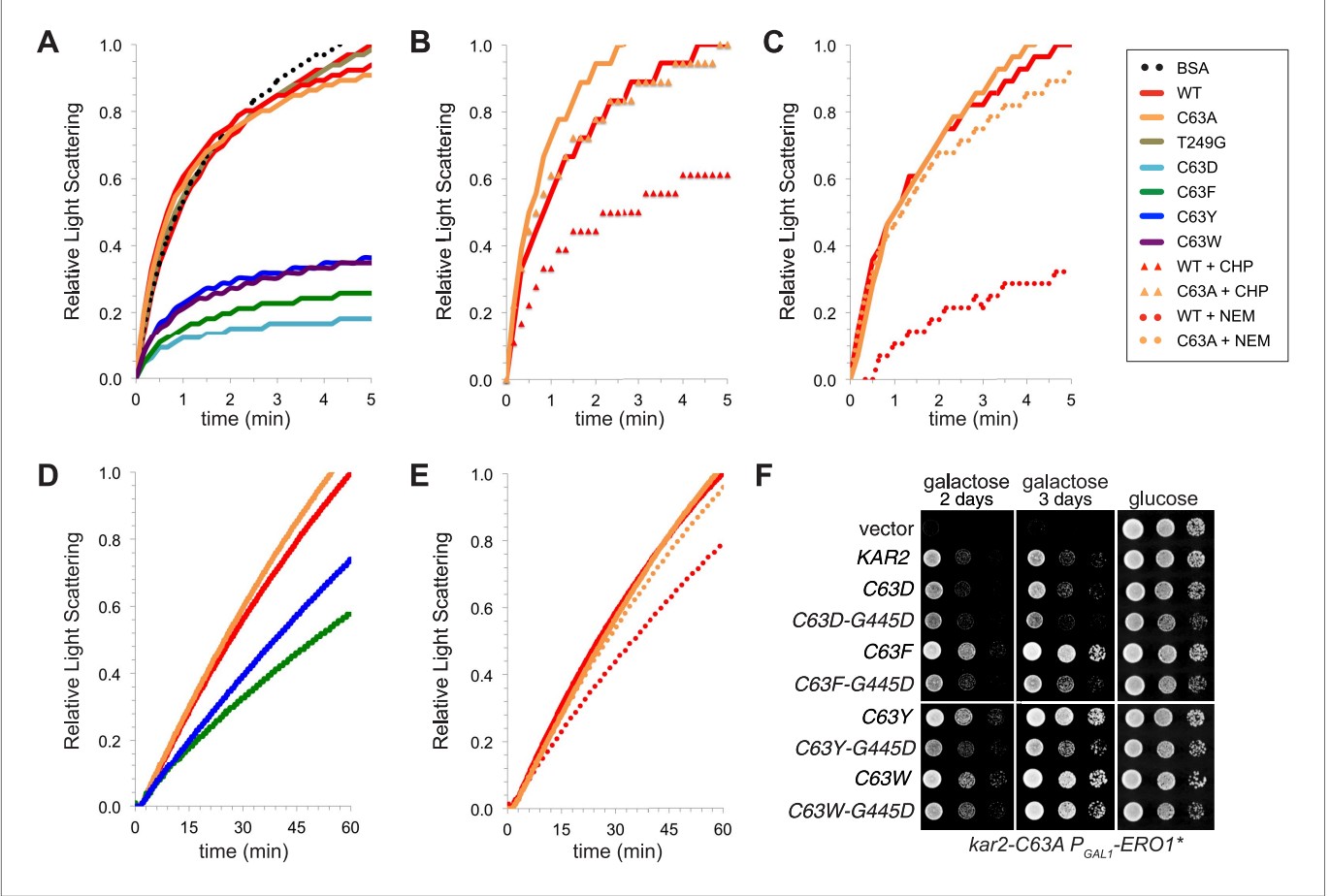

**Figure 8**. BiP cysteine mutants that protect cells during oxidative stress are more effective than wild-type BiP in suppressing polypeptide aggregation. (**A–C**) Denatured rhodanese was diluted to a final concentration of 1 μM in the presence of 4 μM BSA or wild-type, mutant, peroxide-treated, or alkylated BiP. Samples in panels **B** and **C** were mock treated to match the CHP or NEM-treatment. Rhodanese aggregation was followed by monitoring light scattering at 320 nm over a period of 5 min (**D** and **E**) After denaturation in 6 M guanidine and 40 mM DTT, IgY was diluted to final concentration of 0.7 μM at 45°C in the presence or absence of 0.7 μM wild-type, mutant, or alkylated BiP. IgY aggregation was followed by monitoring light scattering at 360 nm over a period of 60 min. Data in panels **A–E** are representative traces from at least three trials. (**F**) CSY278 containing plasmids pCS681, pCS802, pJW5, pCS687, pCS844, pCS688, pCS845, pCS750, pCS846, or empty vector were spotted on SMM-leu or SMM Gal-leu plates, and plates were incubated at 37°C for 2 d (glucose) or 2-3 d (galactose).

wild-type Kar2 (*Figure 8D,E*). Kar2-C63D and Kar2-C63W were prone to aggregation at high temperature (required for IgY aggregation), and we were unable to assess their activity in the IgY assay (data not shown).

These in vitro data suggest that the ability of modified BiP to bind polypeptides could account for the BiP-mediated protection against the deleterious effects of excess ROS in the ER. Consistent with this interpretation, we observed that Kar2 peptide-binding activity contributes to the protective activity of the Kar2-C63D/F/Y/W mutants in cells. A Kar2-C63D/F/Y/W mutant with an additional mutation in the peptide-binding domain, G445D, (analogous to DnaK-G400D; *Burkholder et al., 1996*) showed decreased cell growth under conditions of increased ER ROS (relative to the Kar2-C63D/F/Y/W mutant alone) (*Figure 8F*), indicating that the ability to bind polypeptides is essential for BiP-mediated protection of cells during oxidative stress.

## Discussion

We have demonstrated an additional and new role for the molecular chaperone BiP as a direct sensor of small molecule redox changes in the ER lumen. We observed that under conditions of increased ER

peroxide production (Ero1* overexpression), a conserved cysteine thiol in BiP is oxidized by peroxide and converted to sulfenic acid (*Figure 3*). Biochemical assays of oxidized and alkylated BiP, as well as assays of BiP cysteine-substitution alleles that mimic oxidized BiP, indicate that modification of the BiP cysteine alters BiP chaperone activity, such that ATPase activity is inhibited while avidity of binding to misfolded proteins is increased. Due to the striking conservation of the BiP cysteine among orthologs, we propose that BiP oxidation may be a conserved mechanism for promoting cell viability under conditions of excess ROS within the ER.

## Modulation of chaperone activity during oxidative stress

We propose that BiP oxidation is part of a redox signaling circuit within the ER, wherein hyper-oxidizing conditions in the ER are offset through modulation of BiP activity. A primary outcome of oxidative stress is oxidative damage to proteins, including the mispairing of cysteine residues to form non-native disulfide bonds as well as the oxidation of cysteine thiols with small molecules, which prevents disulfide bond formation. These oxidized proteins are unlikely to be able to fold properly until ER conditions are returned to a less oxidized state. We suggest that activation of a redox switch in BiP upon ROS accumulation in the ER converts BiP from a chaperone driven by ATP hydrolysis to a high avidity polypeptide holdase. An enhanced capacity for oxidized BiP to bind polypeptides may serve to maintain proteins in a folding-competent state until the ER returns to a more reduced status. Upon restoration of a more reduced ER environment, reduction of BiP would reestablish BiP activity as an ATP-driven chaperone to facilitate protein folding through its binding and release of polypeptides. Inactivation of ATPase activity for a pool of BiP upon oxidation may additionally benefit cells by limiting ATP turnover during oxidative stress. In the absence of modification, BiP has the potential to consume ATP while carrying out futile chaperone cycles of binding and release from polypeptides locked into non-native states by covalent bonds; inactivation of BiP ATPase activity would limit hydrolysis of cellular ATP as a byproduct of futile folding cycles. Finally, inhibition of the translocation activity of BiP under hyper-oxidizing conditions could decrease the flux of nascent polypeptides into the ER further decreasing the burden on the ER folding machinery (*Figure 6*). We are unable to definitively link translocation attenuation with BiP modification due to the lack of a measurable translocation defect in wild-type cells under ER oxidative stress. Yet these data do not preclude a more modest decrease in polypeptide flux during oxidative stress; note we do not anticipate all BiP becomes modified during oxidative stress, which is consistent with the lack of a complete translocation block during stress.

Data from our BiP cysteine-mutant alleles suggest that a single population of all reduced or all oxidized BiP is insufficient to maintain cell viability throughout a range of redox conditions. Alleles of BiP that mimic the phenotypic outcomes of BiP oxidation (Kar2-C63D/F/Y/W) are unable to function effectively as the only copy of BiP in cells during standard growth conditions (*Figure 5*). Conversely, a mimetic of reduced BiP (Kar2-C63A) is unable to support robust growth of cells as the only form of cellular BiP during oxidative ER stress (*Figure 2*). We anticipate the robust growth we observed in the presence of both Kar2-C63A plus Kar2-C63F/Y/W is due to the genetic recapitulation of the normally reduced and oxidized pools of ER BiP.

Intriguingly, multiple means may exist to post-translationally control BiP activity to benefit cells under oxidative stress. Recently mammalian BiP was established as a main player in an oxidative stress response pathway initiated by activation (oxidation) of GPx7 (NPGPx) (*Wei et al., 2012*). Wei et al. reported that GPx7 acts as a direct sensor of cellular ROS, and they propose that oxidized GPx7 catalyzes the formation of an intramolecular BiP Cys41-Cys420 disulfide. Similar to what we observed upon BiP oxidation, Wei et al. show that formation of an intramolecular BiP disulfide enhances the ability of BiP to bind denatured luciferase in vitro. Notably, GPx7 and the disulfide-bonded cysteines in BiP are found only in mammals and not in other eukaryotes (yeast BiP has one cysteine), suggesting the GPx7-BiP signaling pathway is likely unique to mammalian cells. Yet it is significant that in their study Wei et al. observed in vitro treatment of recombinant mammalian BiP with peroxide facilitated not only the Cys41-Cys420 disulfide but also a sulfonic acid ($-SO_3H$) at Cys41, which corresponds to Kar2 Cys63 (*Wei et al., 2012*). These data demonstrate that mammalian BiP Cys41 is prone to direct oxidation by peroxide like yeast BiP, solidifying the capacity for BiP to serve also as a direct sensor of ROS in mammalian cells.

Emerging data suggest that decoupling of ATPase and chaperone functions through cysteine oxidation could be a property shared by several Hsp70s. For example, in vitro alkylation of the three

cysteines in the cytosolic yeast Hsp70 Ssa1 triggers a loss of ATPase activity and augments the capacity of Ssa1 to prevent aggregation of denatured luciferase (*Liu et al., 1996*; *Hermawan and Chirico, 1999*). Cys15 in Ssa1 is equivalent to the conserved BiP cysteine, and changes in chaperone activity for alkylated Ssa1 could occur through a mechanism similar to what we describe for BiP. Alternatively, the mechanism may be distinct from what we describe for BiP and make use of the other two Ssa1 cysteine residues; indeed, precedent exists for a redox-sensing role for Ssa1 Cys264 and Cys303 (but not Cys15) in the derepression of Hsf1 in response to a variety of thiol-reactive compounds (*Wang et al., 2012*). Irreversible inhibition of the Hsp70 ATPase activity of mammalian Hsp70 (HSPA1A) has been shown to occur upon oxidation of the same two cysteines with methylene blue (*Miyata et al., 2012*). Intriguingly, a separate study reported that the in vitro acquisition of peptides by cytosolic Hsp70 is enhanced in the presence of hydrogen peroxide, raising the possibility that ATPase and chaperone activities of Hsp70 may also be decoupled by oxidation (*Callahan et al., 2002*). To the best of our knowledge our study shows the first example of an Hsp70 modified by endogenous ROS; given the reported susceptibility of numerous Hsp70s to oxidation by exogenous oxidants it will be exciting to determine if and what endogenous oxidants are sensed by the cytoplasmic and mitochondrial Hsp70s.

Notably, redox modification of Hsp70 cysteines is not restricted to eukaryotes but also has been shown in bacteria. Oxidation of Cys15 in the bacterial Hsp70 DnaK (equivalent to Kar2 Cy63) has been demonstrated to occur when bacteria are exposed to exogenous oxidants at elevated temperatures (*Winter et al., 2005*). Markedly distinct from what we observe for BiP, oxidant exposure inhibits DnaK's ability to prevent luciferase aggregation (*Winter et al., 2005*). Although modification of the DnaK cysteine is detected in cells exposed to oxidant, a DnaK-C15A mutant shows the same oxidant-induced loss of activity as wild-type DnaK, suggesting that cysteine oxidation may contribute to but does not account for the inactivation of DnaK chaperone activity (*Winter et al., 2005*). Interestingly, the inactivation of DnaK during oxidative stress occurs concurrently with oxidant-induced activation of holdase activity for the general chaperone Hsp33 (*Winter et al., 2005*). Thus oxidative stress appears to evoke similar functional outcomes in both bacteria and eukaryotes (loss of Hsp70 ATP-dependent chaperone activity and enhanced holdase capacity), but in eukaryotes this can be achieved through a switch in BiP activity alone whereas in bacteria this outcome is a product of altered activities for a coupled DnaK-Hsp33 system.

## Physical alterations in BiP upon oxidation

The position of the BiP cysteine within the ATP binding pocket (less than 10 Å from the nucleotide) (*Wisniewska et al., 2010*; *Yan et al., 2011*) suggests that oxidation of the cysteine thiol may prevent the correct positioning of the nucleotide within the ATP-binding pocket. Sulfenic acid at the site of the cysteine is unlikely itself to physically block ATP entry into the cleft; it seems more likely that cysteine oxidation will perturb the surrounding residues, which in turn may alter residues necessary for ATP positioning or hydrolysis, leading to decreased ATPase activity. It is well established that the nucleotide and peptide binding domains of BiP are allosterically coupled; nucleotide binding and hydrolysis modulates both the structure of the peptide binding domain and its affinity for polypeptides (*Zuiderweg et al., 2013*). We propose that structural changes within the nucleotide-binding pocket upon cysteine oxidation are similarly propagated to the peptide-binding domain to alter peptide affinity, allowing for enhanced peptide binding by BiP without ATP hydrolysis.

## BiP oxidation adduct(s)

Sulfenic acids are generally considered reactive species prone to further oxidation (*Jacob et al., 2006*; *Reddie and Carroll, 2008*; *Roos and Messens, 2011*), which raises the possibility that direct modification of BiP with peroxide primes BiP for further oxidative modification(s). Our data indicate that addition of sulfenic acid accounts for at least some portion of the oxidized BiP cysteine that occurs upon oxidative stress. However, our data do not rule out the possibility of additional BiP cysteine modifications. Indeed, the sensitivity of the *kar2-C63A* strain to the thiol-oxidant diamide (*Figure 2*), thought to primarily impact the redox balance of cells through oxidation of the cellular anti-oxidant glutathione (*Kosower et al., 1969*), likely indicates the potential for a non-peroxide adduct on the BiP cysteine. Our experiments detect only a small fraction of BiP isolated from cells in an oxidized state; we estimate recovery of less than 1% of the total isolated BiP in a modified form. Low yields of oxidized cellular BiP species have limited our ability to use mass spectrometry for conclusive

identification of any additional modifications. It is worth noting that the limited recovery of oxidized BiP from cells likely reflects the difficulty in preserving oxidized thiols, which are prone to reduction during lysate preparation. We speculate that a pool of modified BiP, greater than the recovered 1%, exists in cells during oxidative stress; indeed, it is difficult to envision that a 1% pool of oxidized BiP in wild-type cells during oxidative stress could account for the striking growth difference between the wild-type and *kar2-C63A* strains during oxidative ER stress (*Figure 2*).

If BiP undergoes oxidation by more than one type of molecule, it is possible that distinct biochemical activities for BiP may be observed dependent on the specific redox modification. Sulfenic acid is known to prime proteins for modification by glutathione (*Gallogly and Mieyal, 2007*), and the abundance of reduced glutathione in the ER could allow for formation of a reversible BiP protein-glutathione adduct. Consistent with the proposed susceptibility of BiP to glutathione modification, a proteomics approach previously identified mammalian BiP (and 22 additional proteins) as a substrate for glutathiolation during diamide-induced oxidative stress in endothelial cells (*Lind et al., 2002*). The efficacy of the BiP-C63F/Y/W alleles may reflect the ability of these larger amino acid side chains to mimic BiP glutathiolation and protection. Sulfenic acid can also be further oxidized by peroxide to form sulfinic ($-SO_2H$) and sulfonic ($-SO_3H$) acid species, which are generally considered irreversible modifications. Modification of BiP by excess peroxide could denote irreparable cellular damage, initiating a pathway for self-destruction (e.g., like apoptotic death induced by unresolved signaling through the unfolded protein response) (*Walter and Ron, 2011*). Conversely, glutathiolation of BiP could be an important means to protect BiP from irreversible oxidation with excess peroxide, mediating a reversible signaling system for stress protection.

## Concluding remarks

We uncovered BiP oxidation upon genetic manipulation of cells to create hyper-oxidizing ER conditions; the ability to alter the redox environment of the ER was key to reveal BiP oxidation and has been similarly vital for identification of other homeostatic pathways in the ER such as the UPR (*Kozutsumi et al., 1988*; *Dorner et al., 1990*; *Gething and Sambrook, 1992*). Although the UPR was originally designated as a stress response system based on its importance during acute stress, it is now appreciated that the UPR plays crucial roles in maintaining ER plasticity and function under normal growth conditions (*Moore and Hollien, 2012*). Similarly, we propose that BiP oxidation likely influences cell growth and development in the absence of extreme redox imbalance. Notably, modified BiP was detectable even in the absence of Ero1* overproduction (*Figure 3*), consistent with a role for BiP oxidation not only during stress but also during non-stressed growth conditions. Increased ROS content within the ER as a consequence of enhanced secretory capacity could increase the pool of oxidized BiP to help maintain secretory protein dynamics. Assays similar to those described herein should enable future studies of yeast and mammalian BiP oxidation under non-stress conditions.

## Materials and methods

### Plasmid construction

Plasmids used in this study are listed in *Table 1*. All yeast expression plasmids derive from the pRS vector series (*Sikorski and Hieter, 1989*). pCS623 and pCS739 contain a XhoI–XhoI fragment of genomic *KAR2*. Additional yeast Kar2-encoding plasmids contain *KAR2* with ~1 kb each of 5′ and 3′ untranslated sequences flanked with engineered BamHI and SacI restriction sites. Plasmids pCS623 and pCS681 show identical phenotypes in yeast. To create pCS757 and pCS760, a FLAG epitope (DNA sequence: GGAGATTATAAGGATGACGACGATAAGGGT) was inserted by two-step fusion PCR immediately prior to the DNA sequence in *KAR2* encoding for the HDEL retrieval signal. pCS584 was made by placing a ~2.4 kb *CAN1* fragment into pRS316, and then replacing an EcoRI-SpeI piece within *CAN1* with an EcoRI-SpeI fragment from pCS452. A pET21b (EMD Millipore, Billerica, MA) Kar2 plasmid series was made that expresses a C-terminal his$_6$-tagged Kar2 residues 40–668. A pET28a (EMD Millipore) Kar2 plasmid series codes for a N-terminally tagged Kar2 amino acids 42–682. To make pCS675, sequence coding for the Sec63J domain (residues 121–221) was cloned into pGEX-4T-1 (GE Healthcare, UK). The UPRE-*lacZ* reporter pCS852 was generated by destroying the *LEU2* marker in pJC8 (*Cuozzo and Kaiser, 1999*) by MfeI digestion followed by ligation of the cut plasmid backbone. Amino acid substitutions were made by QuikChange site-directed mutagenesis (Stratagene, Santa Clara, CA). All mutations were confirmed by sequencing.

**Table 1.** Plasmids

| Name | Description | Markers | Source |
|---|---|---|---|
| pJC8 | *UPRE-LacZ* reporter | *CEN URA3 LEU2* | ***Cuozzo and Kaiser, 1999*** |
| pCS852 | *UPRE-LacZ* reporter | *CEN URA3* | This study |
| pAF112 | $P_{GAL1}$-*ERO1-myc* | *CEN URA3* | ***Sevier et al., 2007*** |
| pCS452 | $P_{GAL1}$-*ERO1\*-myc* | *CEN URA3* | ***Sevier et al., 2007*** |
| pCS504 | $P_{GAL1}$-*ero1\*-C100A-C105A-myc* | *CEN URA3* | ***Sevier et al., 2007*** |
| pCS584 | *can1::*$P_{GAL1}$-*ERO1\*-myc* | *CEN URA3* | This study |
| pCS739 | *kar2-C63A* | *URA3* | This study |
| pCS623 | *KAR2* | *CEN URA3* | This study |
| pCS681 | *KAR2* | *CEN LEU2* | This study |
| pCS685 | *kar2-C63A* | *CEN LEU2* | This study |
| pCS802 | *kar2-C63D* | *CEN LEU2* | This study |
| pCS687 | *kar2-C63F* | *CEN LEU2* | This study |
| pCS688 | *kar2-C63Y* | *CEN LEU2* | This study |
| pCS750 | *kar2-C63W* | *CEN LEU2* | This study |
| pCS774 | *kar2-T249G* | *CEN LEU2* | This study |
| pJW5 | *kar2-C63D-G445D* | *CEN LEU2* | This study |
| pCS844 | *kar2-C63F-G445D* | *CEN LEU2* | This study |
| pCS845 | *kar2-C63Y-G445D* | *CEN LEU2* | This study |
| pCS846 | *kar2-C63W-G445D* | *CEN LEU2* | This study |
| pCS757 | *KAR2-FLAG* | *CEN LEU2* | This study |
| pCS760 | *kar2-C63A-FLAG* | *CEN LEU2* | This study |
| pCS630 | *kar2-(40-668)-His$_6$* | AMP | This study |
| pCS631 | *kar2-(40-668)-C63A-His$_6$* | AMP | This study |
| pJW4 | *kar2-(40-668)-C63D-His$_6$* | AMP | This study |
| pCS658 | *kar2-(40-668)-C63F-His$_6$* | AMP | This study |
| pCS643 | *kar2-(40-668)-C63Y-His$_6$* | AMP | This study |
| pCS644 | *kar2-(40-668)-C63W-His$_6$* | AMP | This study |
| pCS639 | *kar2-(40-668)-T249G-His$_6$* | AMP | This study |
| pCS675 | *GST-sec63J-(121-221)* | AMP | This study |
| pCS817 | *His$_6$-kar2-(42-682)* | KAN | This study |
| pCS818 | *His$_6$-kar2-(42-682)-C63A* | KAN | This study |
| pCS822 | *His$_6$-kar2-(42-682)-C63D* | KAN | This study |
| pCS819 | *His$_6$-kar2-(42-682)-C63F* | KAN | This study |
| pCS820 | *His$_6$-kar2-(42-682)-C63Y* | KAN | This study |
| pCS821 | *His$_6$-kar2-(42-682)-C63W* | KAN | This study |
| pKP113 | *His$_6$-kar2-(42-682)-T249G* | KAN | This study |

## Strains and growth conditions

*Saccharomyces cerevisiae* strains were grown and genetically manipulated using standard techniques (***Adams et al., 1998***). YPD is rich medium with 2% glucose. SMM is synthetic minimal medium supplemented with 2% glucose or a specified carbon source: 2% raffinose (SMM Raf) or 2% galactose (SMM Gal). SCAA is minimal medium containing 0.67% yeast nitrogen base, 2% casamino acids, an amino acid supplement (0.004% adenine, histidine, and methionine, 0.006% leucine and lysine, 0.002% tryptophan), and a specified carbon source: 2% raffinose (SCAA Raf) or 2% galactose (SCAA Gal). Uracil or leucine medium supplements were removed to select for plasmids as needed.

Strains used in this study are listed in *Table 2*. CKY1026 is described in (*Sevier et al., 2007*). The KanMX marker in CKY1026 was swapped for NatMX using homologous recombination to make CSY158 (*Goldstein and McCusker, 1999*). For CSY172, a KanMX-*GAL* promoter module was inserted before the *SUC2* open reading frame using PCR-mediated gene-modification (*Longtine et al., 1998*). To create CSY170, a disrupted *CAN1* fragment containing $P_{GAL1}$-*ERO1\*-myc* was released from pCS584, and transformed into CKY263. Stable integrants were selected on plates lacking arginine and containing 60 µg/ml canavanine. A *kar2Δ* strain (CSY214) was made by replacing the *KAR2* coding sequence with KanMX in a homozygous *GAL2 ura3-52 leu2-3,112* diploid. The resultant heterozygous diploid was transformed with pCS623 and a viable *MATa* Ura+ KanMX+ segregant was recovered after sporulation. CSY289, 290, 368, 292 and 293 were generated by transformation of CSY214 with pCS681, pCS685, pCS802, pCS687, or pCS688, followed by selection against pCS623 by plating on SMM with 5-FOA. CSY308 was made by one-step gene replacement of *PEP4* with NatMX in CSY214 (*Kozutsumi et al., 1988*). CSY316 and CSY319 were generated by transformation of CSY308 with pCS757 or pCS760, followed by counter-selection of pCS623 on SMM with 5-FOA. CSY275 was created by replacement of *KAR2* with *kar2-C63A* using a two-step method with linearized pCS739 as described in *Rothstein (1991)*. Double mutant combinations of the *kar2-C63A, ire1Δ,* and *can1::$P_{GAL1}$-ERO1\*-myc* mutants were created using standard genetic techniques of mating, sporulation, and scoring.

## Recombinant protein purification

BL21 (DE3) pLysS cells containing pET-derived plasmids were grown overnight at 37°C to saturation in Luria–Bertani (LB) medium containing 34 µg/ml chloramphenicol and 100 µg/ml ampicillin or 15 µg/ml kanamycin. Cells were diluted 1:20 in LB with antibiotics, grown for 2 hr at 37°C, moved to 24°C, and protein expression was induced with 0.4 mM isopropyl-β-D-thiogalactopyranoside (IPTG). Cells were harvested 2–6 hr post-IPTG addition, and cell pellets were frozen at −80°C. Pellets were thawed and solubilized for 30 min on ice with 20 ml sonication buffer (50 mM HEPES, pH 7.4, 0.3 M NaCl, 10 mM imidazole) plus 1 mM PMSF, 1 µM pepstatin A, and 5 mM BME per 1 l cell culture. Cells were lysed by treatment with lysozyme followed by sonication, and insoluble material was removed by centrifugation at 16,000×*g* for 20 min. Supernatant was incubated for 30 min at 4°C with a slurry of 50% Ni-NTA agarose resin (Qiagen, Germany) and loaded into a column, or lysate was loaded directly onto a HiTrap

**Table 2.** Yeast strains

| Strain | Genotype | Source |
| --- | --- | --- |
| CKY263/CSY5 | *MATa GAL2 ura3-52 leu2-3,112* | Lab collections |
| CKY264/CSY6 | *MATα GAL2 ura3-52 leu2-3,112* | Lab collections |
| CKY1026/CSY44 | *MATa GAL2 ura3-52 leu2-3,112 ire1Δ::KanMX* | Lab collections |
| CSY158 | *MATa GAL2 ura3-52 leu2-3,112 ire1Δ::NatMX* | This study |
| CSY172 | *MATa GAL2 ura3-52 leu2-3,112 ire1Δ::NatMX KanMX:$P_{GAL1}$-SUC2* | This study |
| CSY275 | *MATa GAL2 ura3-52 leu2-3,112 kar2-C63A* | This study |
| CSY277 | *MATa GAL2 ura3-52 leu2-3,112 kar2-C63A ire1Δ::NatMX* | This study |
| CSY170 | *MATa GAL2 ura3-52 leu2-3,112 can1::$P_{GAL1}$-ERO1\*-myc* | This study |
| CSY278 | *MATa GAL2 ura3-52 leu2-3,112 kar2-C63A can1::$P_{GAL1}$-ERO1\*-myc* | This study |
| CSY214 | *MATa GAL2 ura3-52 leu2-3,112 kar2Δ::KanMX* [pCS623] | This study |
| CSY289 | *MATa GAL2 ura3-52 leu2-3,112 kar2Δ::KanMX* [pCS681] | This study |
| CSY290 | *MATa GAL2 ura3-52 leu2-3,112 kar2Δ::KanMX* [pCS685] | This study |
| CSY368 | *MATa GAL2 ura3-52 leu2-3,112 kar2Δ::KanMX* [pCS802] | This study |
| CSY292 | *MATa GAL2 ura3-52 leu2-3,112 kar2Δ::KanMX* [pCS687] | This study |
| CSY293 | *MATa GAL2 ura3-52 leu2-3,112 kar2Δ::KanMX* [pCS688] | This study |
| CSY308 | *MATa GAL2 ura3-52 leu2-3,112 kar2Δ::KanMX pep4Δ::NatMX* [pCS623] | This study |
| CSY316 | *MATa GAL2 ura3-52 leu2-3,112 kar2Δ::KanMX pep4Δ::NatMX* [pCS757] | This study |
| CSY319 | *MATa GAL2 ura3-52 leu2-3,112 kar2Δ::KanMX pep4Δ::NatMX* [pCS760] | This study |

chelating column (GE Healthcare) charged with nickel. C-terminally tagged Kar2 preps were washed extensively to remove contaminating ATPase activity similar to (*McClellan et al., 1998*). Briefly, resin was washed with 10 column volumes (cv) sonication buffer, 10 cv sonication buffer with 5% glycerol, 1% Triton-X-100, 10 cv sonication buffer with 5% glycerol, 1 M NaCl, 10 cv sonication buffer with 5% glycerol, 5 mM ATP, 10 mM MgCl$_2$, 10 cv sonication buffer with 5% glycerol, 0.5 M Tris–HCl, pH 7.4, and 10 cv sonication buffer with 5% glycerol, 25 mM imidazole. Purified protein was eluted with 3 cv sonication buffer with 5% glycerol, 0.25 M imidazole. Protein was exchanged into 40 mM Tris–HCl, pH 7.4, 80 mM NaCl, 10% glycerol using a PD-10 column. For N-terminally tagged Kar2, resin was washed with 5 cv wash buffer (20 mM HEPES, pH 7.5, 0.5 M NaCl, 10% glycerol, 10 mM imidazole) and 15 cv wash buffer with 25 mM imidazole. Purified protein was eluted with 3 cv wash buffer with 0.2 M imidazole. Protein was exchanged into 10 mM Tris–HCl, pH 7.4, 50 mM NaCl, 10% glycerol using a NAP-5 column. All rhodanese and IgY assays were performed with N-terminally tagged Kar2; the C-terminally tagged Kar2 is not full length and was not active in the protein aggregation assays.

GST-Sec63J protein (pCS675) was expressed in EN2 cells (*dnaKΔ*), which were generously provided by Nadia Benaroudj (Institut Pasteur, Paris, France) and are described in *Ratelade et al. (2009)*. Bacteria containing pCS675 were grown overnight to saturation at 30°C in LB medium containing 100 µg/ml ampicillin, diluted 1:20 in LB with ampicillin, grown for 2.5 hr at 30°C, and protein expression was induced with 0.2 mM IPTG. Cells were harvested 2 hr post-IPTG addition and cell pellets were frozen at −80°C. Pellets were thawed and solubilized for 30 min on ice with 20 ml PBS plus 2 mM EDTA, a complete protease inhibitor cocktail (Roche, Switzerland), 1 mM PMSF, and 5 mM BME per 1 l cell culture. Cells were lysed by treatment with lysozyme followed by sonication. After addition of benzonase and 0.1% Triton-X-100, insoluble material was removed by centrifugation at 20,000×*g* for 20 min. Supernatant was incubated for 1 hr at 4°C with a slurry of 50% glutathione-sepharose (GE Healthcare). Resin was loaded into a column and washed with 20 cv PBS with 2 mM EDTA, 20 cv PBS with 2 mM EDTA, 1 M KCl, 0.1% Triton-X-100, and 10 cv PBS. Protein was eluted with 10 cv 50 mM Tris–HCl, pH 8, 10 mM reduced glutathione, 10% glycerol. A vivaspin-15 column (GE Healthcare) was used for glutathione removal and buffer exchange into 20 mM HEPES, pH 6.8, 75 mM KOAc, 0.25 M sorbitol, 5 mM MgOAc$_2$, 10% glycerol.

All purified proteins were flash frozen in liquid nitrogen and stored at −80°C. Protein concentrations were determined by BCA protein assay (Thermo Fisher Scientific, Waltham, MA) using bovine serum albumin as a standard.

To prepare alkylated Kar2, 1 nmol Kar2 was incubated with 20 nmol TCEP (diluted from a 0.1 M TCEP stock prepared in 1.5 M Tris–HCl, pH 8.8) for 1 hr at room temperature in a 40 µl total volume of buffer suitable for downstream assays. Kar2 was then incubated with 40 nmol NEM for 1 hr at room temperature, and the reaction was quenched with 1 µmol DTT. To prepare oxidized Kar2, 1 nmol Kar2 was incubated with 70 nmol CHP for 30 min at 30°C in 70 µl total volume of 10 mM Tris-HCl, pH 7.4, 50 mM NaCl. CHP was subsequently removed using a Bio-spin P6 column (Bio-Rad, Hercules, CA) equilibrated with 10 mM Tris-HCl, pH 7.4, 50 mM NaCl. Unmodified Kar2 control proteins were prepared identically except buffer was substituted for NEM or CHP.

## ATPase activity

To measure ATPase activity, 1 µM C-terminally tagged Kar2 and 2.5 µM GST-Sec63J were incubated with 0.1 mM of cold ATP and 0.45 µCi of [alpha-$^{32}$P]ATP (Perkin–Elmer, Waltham, MA) in ATPase buffer (50 mM Tris–HCl, pH 7.4, 50 mM KCl, 5 mM MgCl$_2$, 1 mM DTT) in a total volume of 45 µl at room temperature. For peroxide-treated Kar2 samples, 3 µM Kar2 and 5 µM GST-Sec63J were incubated in assay buffer lacking DTT (to prevent Kar2 reduction). At various time points, 5 µl aliquots were removed and activity was quenched with the addition of an equal volume of 2X stop solution (0.14% SDS, 32 mM EDTA, 0.2 M NaCl). Samples (1–2 µl) were spotted on polyethyleneimine cellulose TLC plates (Sigma-Aldrich, St. Louis, MO), and plates were developed in 1 M formic acid and 0.5 M LiCl. Conversion of ATP to ADP was imaged with a phosphorimager and quantified using ImageQuant (GE Healthcare).

## Unfolded protein response detection

Strains transformed with the UPRE-*lacZ* reporter plasmid pJC8 (*Cuozzo and Kaiser, 1999*) or pCS852 were grown in SMM medium and treated with 0 or 2 mM DTT for 1.5 hr (37°C) or 2 hr (30°C) prior to harvesting of exponential phase cells. Cells were permeabilized and beta–galactosidase activity measured as described (*Guarente, 1983*). Three transformants were assayed in duplicate per strain.

## Translocation westerns

CSY289, 290, 368, 292, and 293 were cultured in YPD at 24°C until mid-log phase, at which time samples were divided and half of the cultures were moved to a 37°C water bath incubator. After 90 min, 5 OD$_{600}$ units were harvested for each culture. CSY44 and CSY172 containing pAF112, pCS452, or empty vector were grown overnight at 30°C in SMM Raf or SMM with 2% lactate and 2% gycerol, respectively. Cells were subcultured into SMM Gal and 5 OD$_{600}$ equivalents were harvested after 5 hr of growth at 30°C. Lysates were prepared as described (*Kushnirov, 2000*). Samples were suspended in 100 µl of sample buffer (62.5 mM Tris–HCl, pH 6.8, 10% glycerol, 2% SDS, 0.01% bromophenol blue) containing 2% BME and 10 µg/ml pepstatin A and boiled for 3 min at 100°C. Proteins were separated by SDS-PAGE and detected with appropriate antiserum after transfer to nitrocellulose.

## Biotin-switch assay

CSY316 and CSY319 containing pRS316 or pCS452 were cultured overnight at 30°C in SMM Raf or SCAA Raf, subcultured into SMM Gal or SCAA Gal the next morning, and grown for 6–8 hr at 30°C. Cells (10 OD$_{600}$ equivalents) were harvested by centrifugation, and pellets were flash frozen in liquid nitrogen and stored at −80°C. Cells were suspended in 40 µl of 10% TCA. Zirconium beads were added, and cells were lysed in a FastPrep 24 instrument (MP Biomedical, Santa Ana, CA) with two 1 min pulses at speed 6 separated by a 5 min rest on ice. Samples were diluted with 1 ml of 10% TCA, and liquid was transferred to a new tube. Proteins were precipitated by centrifugation at 21,000×*g* for 10 min at 4°C, and pellets were washed once with ice-cold 5% TCA and once with ice-cold ethanol. Pellets were suspended in 500 µl of urea-containing cysteine modification buffer (CMBU) (0.1 M HEPES-NaOH, pH 7.4, 1% SDS, 10 mM DTPA, 6 M urea) with a complete ultra protease inhibitor cocktail (Roche) and 0.1 M NEM. Samples were rotated for 30 min at room temperature, 500 µl of CMBU with or without 10% BME was added, and samples were rotated for another 15 min at room temperature. Proteins were separated from small molecules by centrifugation after a 5 min incubation on ice with 10% TCA as above. Pellets were washed once with 5% TCA, twice with ethanol, and suspended in 300 µl of CMBU with 0.1 mM maleimide-biotin (Sigma-Aldrich). For sodium arsenite treatment, pellets previously untreated with BME were suspended in 300 µl of CMBU with 20 mM sodium arsenite (Sigma-Aldrich) and 0.1 mM maleimide-biotin. After a 30 min rotation at room temperature, unreacted maleimide was quenched with 4% BME for 5 min, and small molecules and proteins were separated by addition of TCA as above. Pellets were solubilized in 100 µl CMBU, diluted with 1 ml IP buffer (50 mM Tris–HCl, pH 7.4, 0.15 M NaCl, 1% Triton-X-100), and incubated for 10 min at room temperature. Insoluble material was removed by centrifugation at 21,000×*g* for 5 min at 4°C prior to addition of 30 µl of 50% anti-FLAG M2 bead slurry (Sigma-Aldrich). Samples were rotated at 4°C for 1 hr, beads were washed three times with IP buffer, and proteins were eluted for 5 min at 100°C with 30 µl 2X sample buffer containing 10% BME. Proteins were separated by SDS-PAGE, transferred to nitrocellulose, and probed with an avidin-Alexa488 conjugate or Kar2 antiserum and an Alexa546-conjugated anti-rabbit IgG. Fluorescent signal was detected using a ChemiDoc MP System (Bio-Rad).

## DAz-2 modification

For Ero1* experiments, CSY316 and CSY319 containing pRS316 or pCS452 were grown overnight at 30°C in SMM Raf, subcultured the following morning in SMM Gal, and returned to 30°C for 4 hr. For CHP treated samples, cells were grown in SMM at 30°C to mid-log at which time cells were treated with 5 mM CHP for 30 min. Cells (20 OD$_{600}$ equivalents) were harvested by filtration or centrifugation and suspended in 100 µl of lysis buffer (0.1 M HEPES, pH 7.4, 0.3 M NaCl, 10% glycerol, 0.1 mM DTPA) with 1 mM PMSF, 200 U/ml catalase, 2 mM DAz-2 (Cayman Chemicals, Ann Arbor, MI), and 2X complete protease inhibitor cocktail. Zirconium beads were added, cells were lysed in a FastPrep 24 instrument with three 1 min pulses at speed 4 separated by 5 min rests on ice, and 100 µl of lysis buffer with 1 mM DAz-2 and 10% Triton-X-100 was added prior to sample rotation for 20–40 min at room temperature. Lysis buffer (800 µl) was added, samples were incubated for 20 min at room temperature, and insoluble material was removed by centrifugation at 14,000×*g* for 5 min. Anti-FLAG M2 beads (50 µl of a 50% slurry) were added to the soluble material and incubated for 1 hr at 4°C. Beads were washed twice with lysis buffer containing 1% Triton-X-100, and once with lysis buffer. Beads were incubated with 25 µl of lysis buffer without DTPA and with 100 µM phosphine-biotin (Cayman Chemicals) for 2 hr at 37°C, and the reaction was quenched with 25 µl of 2X sample buffer. Proteins were separated by SDS-PAGE, transferred to nitrocellulose, and probed with an avidin-Alexa488 conjugate or Kar2

antiserum and a fluorescent or HRP-conjugated secondary antibody. Fluorescent and chemiluminescent signals were detected using a ChemiDoc MP System.

## Protein aggregation assays

Denatured rhodanese was prepared by incubating 50 µM bovine rhodanese (Sigma-Aldrich) for 1 hr at room temperature in 20 mM HEPES, pH 7.4, 6 M guanidine-HCl, 0.1 M NaCl, 5 mM DTT. Denatured protein aliquots were flash frozen and stored at −80°C. Rhodanese aggregation was performed by diluting thawed denatured rhodanese to 1 µM in 0.2 ml of 20 mM HEPES, pH 7.4, 50 mM KCl containing 5 mM $MgCl_2$, 1 mM ATP, and 4 µM Kar2 or 4 µM BSA. Rhodanese aggregation was followed by monitoring the light scattering at 320 nm for 5 min. IgY was purified using the Chicken IgY Purification kit (Thermo Fisher Scientific) and suspended in 0.1 M Tris–HCl, pH 8, at a final concentration of 50 mg/ml. IgY aggregation was performed according to the methods of *Stronge et al. (2001)* with 0.7 µM final of each Kar2 and IgY. IgY was denatured for a minimum of 2 hr, and used within 3 hr of preparation.

## Acknowledgements

This work was supported by Cornell University, grants from the National Institutes of Health (GM46941) to CAK and (GM105858) to CSS, and a National Science Foundation Graduate Research Fellowship to KAP.

## Additional information

### Funding

| Funder | Grant reference number | Author |
| --- | --- | --- |
| National Institute of General Medical Sciences | GM105858 | Carolyn S Sevier |
| Cornell University | Startup funds | Carolyn S Sevier |
| National Science Foundation | NSF GRF | Kristeen A Pareja |
| National Institute of General Medical Sciences | GM46941 | Chris A Kaiser |

The funders had no role in study design, data collection and interpretation, or the decision to submit the work for publication.

### Author contributions

JW, KAP, Reviewed and edited the manuscript, Acquisition of data; CAK, Reviewed and edited the manuscript, Analysis and interpretation of data; CSS, Wrote the manuscript, Conception and design, Acquisition of data, Analysis and interpretation of data

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
