## [Decision Letter]

Thank you for sending your work entitled “Redox signaling via BiP protects cells against endoplasmic reticulum-derived oxidative stress” for consideration at *eLife*. Your article has been favorably evaluated by Randy Schekman (Senior editor) and 3 reviewers, one of whom is a member of our Board of Reviewing Editors.

The Reviewing editor and the other reviewers discussed their comments before we reached this decision, and the Reviewing editor has assembled the following comments to help you prepare a revised submission.

As you will see all three reviewers appreciated the importance of your paper.

All three reviewers agreed that the following three points need to be addressed:

1) Figure 5 reveals a translocation defect at elevated temperature in cells in which the endogenous KAR2 has been deleted and its function substituted for by plasmids encoding mutant BiPs with modified-cysteine-mimetic-mutations (C63D, C63F and C63Y) but not the neutral (C63A) mutation. The authors have presented this as evidence that modification of wildtype BiP on C63 begets a translocation defect (they mimic the modification with the mutation). From this perspective it is expected that the experiment be conducted under conditions in which the counterparts to the modified BiP (that is unmodified WT and C63A mutant BiP) would have no translocation defect and this is how we understand the experiment to have been designed and executed.

Further experimental evidence for the imposition of a translocation defect by the modified BiP would greatly strengthen the manuscript and if you are able to muster such evidence please do so. For example, a comparison between the level of translocation defect in cells with WT BIP versus BiP C63A after they have been subjected to a conditions that trigger the modification on WT BiP (e.g. ERO1* overexpression, diamide). The predicted outcome of this experiment is a less pronounced translocation defect in the BiP C63A cells that cannot actuate a translocation block.

However, we recognize that this goal may not be attainable (for example, if the oxidizing conditions are not associated with a measureable translocation defect in the WT cells in the first place). In such an instance we would ask that you elaborate on this point, explaining to your readers that was an obvious consideration and enumerate the reasons why the experiments could not be done in an informative manner.

2) The stoichiometry of the modification. It seems hard to imagine that a modification that affects 1-2% of the BiP in the cell could have such far-reaching consequences. Please discuss this point in further detail. If you deem it possible that your measurements are underestimating the extent of the modification, kindly incorporate this into the Discussion.

3) Please address the possibility that the benefit provided to the cell by BiP modification is an enhanced UPR. This can be addressed as a discussion point.

The reviewer elaborate on these points as below.

Reviewer 1

This well written paper reports on a detailed study of the responsiveness of yeast BiP to peroxides.

It begins with the demonstration that yeast BiP's single cysteine C63 undergoes a chemically reversible (by DTT) modification in the ER exposed to hyperoxidizing conditions. The importance of this modification is highlighted by the observations that whereas the Kar2_C63A mutation fully covers the kar2 deficiency under normal conditions, the mutant is impaired in growth under the very conditions associated with enhanced rates of oxidative modification of C63. The benefits of modified C63 under these hyperoxidizing conditions can be mimicked by providing in trans alleles of Kar2 with bulky substitutions at residue 63.

These rescuing alleles and the oxidized BiP share important biochemical features *in vitro* in that they are defective in ATP hydrolysis, are unable to cover the kar2Δ but retain constitutive substrate binding activity and serve as better holdases (i.e. hold unfolded proteins better) than the wildtype yeast BiP. The expression of these altered BiP forms results in a conspicuous translocation defect, which, the authors speculate may underlie part of their protective effect against hyperoxidizing conditions (by defending the organelle against excess unfolded protein load).

Beyond the wealth of details relating to a particular circumstance in the stressed ER, this study calls attention to the layers of regulation affecting chaperone function in the ER and highlights the fact that the protein folding machinery in the secretory pathway is responsive to modulation by important post-translational mechanisms. Therefore, while the basis for the benefit of C63 modification in the hyperoxidised ER remains a subject for (intelligent) speculation, the paper makes an important addition to our understanding of the ER as a protein folding compartment.

Specific issues:

It would be nice to round off the characterization of the oxidized BiP with an analysis of its response to ATP: If indeed the mutation uncouples BiP's two domains, one would expect that the addition of ATP would not lead to substrate release.

Reviewer 2

Oxidative protein folding in the ER is potentially a very significant source of damaging cellular reactive oxygen species (ROS). Although enzymes have been identified in higher eukaryotes to aid in their neutralization, they do not appear to exist in lower eukarotes including yeast. In this study a novel mechanism for sensing and responding to hyper-oxidation of the ER is described, which involves oxidation of the single cysteine in Kar2 found in the nucleotide binding domain. Although direct detection of this modification was not possible, convincing data are obtained through employing a thiol-switch assay and a variety of Kar2 mutants. Oxidation of this cysteine dramatically reduces Kar2p ATPase activity as do several “bulky residue” substitutions at this site, which are argued to mimic cysteine modifications. Due to their defective ATPase activity, these mutants cannot substitute for loss of Kar2p but they complement yeast bearing a C63A mutant, which cannot be modified in response to hyper-oxidation of the ER by ERO1*, as well as wild-type Kar2. Similar to other Kar2 ATPase mutants reported previously, these mutants are defective in translocation of protein into the ER at non-permissive temperature, and additional data reveal they are even better at preventing the aggregation of denatured rhodanese that wild-type Kar2, which leads them to conclude that oxidation of Kar2 under hyper-oxidizing conditions results in Kar2 acting as a holdase to prevent aggregation of existing proteins and to inhibit new proteins from entering the ER thus serving a double beneficial role. The ability to modify Kar2 activity, as opposed to simply changing levels in response to a stress is a very intriguing finding and given the fact that the same cysteine in mammalian was recently shown to be reactive allows one to speculate that this mechanism may be conserved for other ER Hsp70s.

Specific points.

1) It is somewhat surprising that the translocation assay in Figure 5 was done using an elevated temperature, conditions under which wild-type Kar2 is not modified as opposed to using the hyper-oxidizing conditions where this model argues wild-type Kar2 would now be defective in translocation.

2) Studies in Figure 8 included a Kar2 ATPase mutant in an attempt to tie the enhanced suppression of rhodanese aggregation to an increased holdase activity in the cysteine substitution mutants. I am convinced this idea is likely to be correct, but the choice of the Kar2 ATPase mutant used might have prevented a clean link to this point. The T249G mutant that was used cannot hydrolyze ATP, which is included in the assay and thus doesn't lock onto the substrate, and the wild-type Kar2 would likely be continuously released. A better choice might be either the G246D or G247D mutants that do not bind nucleotide and thus remain in the substrate-locked position. Both of these have been reported to prevent translocation in yeast. Another choice would be a mutant corresponding to mammalian T37G, which does not undergo the requisite conformation change upon nucleotide binding.

Reviewer 3

Wang et al. investigate the role of a conserved cysteine residue (Cys63) in Bip/Kar2 that is hypothesized to serve as sensor for redox status in the lumen of the ER. When the ER environment was hyperoxidizing, the Cys63 residue in Kar2 was oxidized to sulfenic acid and cells harboring a Kar2 C63A mutation to block this modification were viable but more sensitive to oxidative stress. These results suggested that the sulfenic acid modification could play a protective role against oxidative stress and when bulky amino acids were substituted for this cysteine to mimic the modification, cells became more tolerate to oxidative stress when expressed in a Kar2 C63A background. Interestingly the bulky cysteine substitution alleles decrease Kar2 ATPase activity and produced thermosensitive translocation blocks *in vivo* when expressed as the sole source of Kar2. *In vitro* experiments with purified proteins showed that rhodanese aggregation rates were markedly reduced by mutant versions of Kar2 containing bulky Cys63 substitutions compared to wild type Kar2 or an ATPase dead version. Moreover double mutants that contained bulky Cys63 substitutions and a mutation in the peptide-binding domain of Kar2 no longer protected cells to oxidative stress. Based on these findings the authors propose that Kar2 helps manage oxidative stress through redox conversion of Cys63 to a sulfenic acid modified form of Kar2 that binds folding intermediates to prevent protein aggregation but hydrolyzes ATP at reduced rates to slow polypeptide translocation, protein folding and ATP consumption until a favorable redox state returns.

Overall this is a strong study that has broad implications for understanding how cells can manage reactive oxygen species in the lumen of the early secretory pathway. While the experimental results are consistent with their proposal, there may be alternative explanations for some results and additional concerns are listed below.

1) An alternative explanation for the results could be that conversion of Cys63 in Kar2 to bulky constituents simply induces a UPR that helps manage oxidative stress. This could be due to an increase in unfolded proteins in the ER or less binding of the mutant Kar2 to Ire1. Figure 5 shows that the C63D, C63F and C63Y versions activate the UPR when expressed as the sole copy, and produce a robust response when shifted to 37C. Do cells that express both Kar2 C63A and Kar2 C63F exhibit a higher level of UPR compared to the single Kar2 C63A? If so, this could suppress the increased ER oxidation levels caused by ERO1* observed in Figure 4.

2) A prediction from the model is that GAL1 induced over-expression of ERO1* in the Kar2 C63A mutant would cause an accumulation of unfolded secretory proteins in the ER and/or translocation defects compared to the wild type Kar2 background. Have these phenotypes been examined in the CSY170 and CSY278 strains?

3) Please clarify the percentage of Kar2 that is oxidized at the Cys63 residue when ERO1* is over-expressed. From the results in Figure 3, there is a 1.8 fold increase in the level of BME-reducible modification on Kar2 under the hyper-oxidizing conditions. However in the Discussion it is mentioned that less than 1% of total isolated BiP is in an oxidized state. Does this mean that less than 2% of Kar2 is oxidized when ERO1* is over-expressed? Could this level of modification have a major impact on translocation and folding rates?

---

## [Author Response]

*1)*
Figure 5
*reveals a translocation defect at elevated temperature in cells in which the endogenous KAR2 has been deleted and its function substituted for by plasmids encoding mutant BiPs with modified-cysteine-mimetic-mutations (C63D, C63F and C63Y) but not the neutral (C63A) mutation. The authors have presented this as evidence that modification of wildtype BiP on C63 begets a translocation defect (they mimic the modification with the mutation). From this perspective it is expected that the experiment be conducted under conditions in which the counterparts to the modified BiP (that is unmodified WT and C63A mutant BiP) would have no translocation defect and this is how we understand the experiment to have been designed and executed*.

*Further experimental evidence for the imposition of a translocation defect by the modified BiP would greatly strengthen the manuscript and if you are able to muster such evidence please do so. For example, a comparison between the level of translocation defect in cells with WT BIP versus BiP C63A after they have been subjected to a conditions that trigger the modification on WT BiP (e.g. ERO1* overexpression, diamide). The predicted outcome of this experiment is a less pronounced translocation defect in the BiP C63A cells that cannot actuate a translocation block*.

*However, we recognize that this goal may not be attainable (for example, if the oxidizing conditions are not associated with a measureable translocation defect in the WT cells in the first place). In such an instance we would ask that you elaborate on this point, explaining to your readers that was an obvious consideration and enumerate the reasons why the experiments could not be done in an informative manner*.

We agree that further experimental evidence for the imposition of a translocation defect by modified BiP is necessary to provide a more compelling link between oxidation of BiP and translocation attenuation. Unfortunately, as the reviewers anticipated, this goal has not been attainable; our attempts to solidify the link between modification of BiP and translocation attenuation have not yielded consistent compelling data.

We now enumerate in the text the experiments that we tried and the reasons they have proven uninformative (in the Results and Discussion sections).

*2) The stoichiometry of the modification. It seems hard to imagine that a modification that affects 1-2% of the BiP in the cell could have such far-reaching consequences. Please discuss this point in further detail. If you deem it possible that your measurements are underestimating the extent of the modification, kindly incorporate this into the Discussion*.

We think it is not only possible but also likely that our measurements are underestimating the extent of the modification of BiP. We have incorporated this into the Discussion section.

*3) Please address the possibility that the benefit provided to the cell by BiP modification is an enhanced UPR. This can be addressed as a discussion point*.

We have added a panel to Figure 7 (panel C) that begins to experimentally address the connection between BiP modification and the UPR. In brief, we show that (modest) UPR induction does occur upon expression of the Kar2-C63D/F/Y/W alleles in a kar2-C63A mutant background. We also show that, although UPR induction occurs and may contribute to protection during oxidative stress, UPR induction is not sufficient for protection during oxidative stress (Figure 7). Our data show a Kar2-T249G mutant, which cannot protect cells during stress, exhibits the same (modest) UPR induction as the protective Kar2-C63D/F/Y/W mutants. These data are discussed in the Results section.